# Utilization of Hot-Spring-Water-Bound CO$_2$ for Horticulture Plants Using Incubation Method

**Kyohei Yokota**

Department of Civil Engineering, National Institute of Technology, Wakayama College, Gobo 644-0023, Japan; yokota@wakayama.kosen-ac.jp; Tel.: +81-738-29-8446

**Abstract:** This study focused on free carbon dioxide (H$_2$CO$_3$) contained in volcanic hot spring water. It was clarified whether free carbon dioxide can be emitted into the atmosphere and increase the CO$_2$ concentration in greenhouses by using the incubation method. Factors influencing the increase in CO$_2$ concentration were identified based on implementation time in a demonstration experiment, temperature and humidity in the incubator, the amount of hot spring water, and the contact area between the hot spring water and the air. An incubator with an inner volume of $2.17 \times 10^7$ mm$^3$ was used in this study. The results showed that unrelated factors were the implementation time in the demonstration experiment, temperature, and humidity. There was a relationship with the amount of hot spring water. The increment of CO$_2$ concentration in the air by free carbon dioxide increased proportionally by increasing the amount of water. Free carbon dioxide contained in hot spring water can be utilized by considering the volume of facility horticulture and plant factories. The ideal methods for utility are to replace the hot spring water every few minutes and to increase the contact area between the air and the hot spring water.

**Keywords:** hot spring water; free carbon dioxide; carbon dioxide; facility horticulture; plant factory





## 1. Introduction

The global population is growing, and according to the United Nations Population Fund (UNFPA), the world population exceeded 8 billion in November 2022 [1]. Its population is expected to continue to grow and is projected to exceed 9 billion by 2050 [2]. Food supply and demand is expected to increase accordingly. Therefore, in order to secure food, it is necessary to cultivate many plants and further accelerate their growth. However, the prices of resources for this purpose are skyrocketing worldwide [3]. Therefore, the prices of resource energy and mineral resources used for growing plants are also increasing, making it difficult to secure food. Furthermore, in the context of global warming, emissions of carbon dioxide and other artificial greenhouse gases need to be curbed [4]. This consideration makes it impossible to avoid the economic impact. As a countermeasure, fiscal decentralization has been shown to have the potential to reduce CO$_2$ emissions [5]. From the perspective of the SDGs, plants need to be grown using methods that address these various issues in a sustainable manner [6].

It is generally known that plants are absorbing CO$_2$ during photosynthesis [7]. Therefore, CO$_2$ must be supplied constantly for growing plants [8]. This has led to research on the optimal management of CO$_2$ for many plants, such as tomatoes, cucumbers, miniature roses, rice, eucalyptus, and strawberries [9,10]. Furthermore, there is a relationship between plant growth and CO$_2$ concentration in the air. Increasing the concentration of CO$_2$ causes photosynthesis to be more active. Therefore, plants grow faster when CO$_2$ concentration is high [11,12]. Generally, the rate of photosynthesis increases proportionally with increasing CO$_2$ concentration [13–16]. In several previous studies, plant growth was enhanced by increasing the concentration of CO$_2$. Elevated atmospheric CO$_2$ will increase nitrogen absorption. And if nutrients continue to be supplied, long-term growth will be

maintained [17–22]. Therefore, it is preferable to maintain a high $CO_2$ concentration for growing a plant in the short term [23,24]. A common method of maintaining $CO_2$ concentration is to make and exchange the air in a facility with outside air by using a ventilation fan [25]. According to the IPCC Sixth Assessment Report, $CO_2$ concentration in the air is increasing to 410 ppm as the annual average value due to human activities [4]. Ventilation methods cannot greatly exceed this concentration. A method of supplying $CO_2$ other than ventilation fans is needed for making plants grow faster [26–28]. In a further method of supplying $CO_2$, plants are grown in horticultural agriculture by reusing $CO_2$ emitted when boilers are burned [29,30]. Other methods use liquefied $CO_2$ or biogas for $CO_2$ supply [31]. Organic carbonate soil is the largest terrestrial carbon reservoir for the supply of $CO_2$. According to the review, worldwide, soils release about 10 times more greenhouse gases compared to fossil fuel combustion [32]. The use of these gases is suitable from a carbon-neutral perspective. Many methods have been considered to supply $CO_2$; however, most of them require $CO_2$-emitting energy resources, such as petroleum. According to a report by the IPCC, a 48% reduction in global carbon dioxide emissions in 2030, a 65% reduction in 2035, and an 80% reduction by 2040 are needed to prevent a 1.5 °C temperature increase from pre-industrial times [33]. The best method is one that can supply $CO_2$ as quickly as a ventilation fan and does not require many resources. The possible alternative resource is hot spring water. Many hot spring waters have high temperatures without an addition of energy resources, and many of them have water temperatures above 25 °C. Some hot spring waters contain a lot of free carbon dioxide ($H_2CO_3$). These hot springs are generally called carbonated springs. Carbonated springs are found throughout the world, including in California, Italy, the Netherlands, and Japan [34–37]. The effect of carbonated spring water on plant growth was verified [38]. The direct effect of the increased $CO_2$ concentration greatly improved the utilization efficiency of water and nitrogen. Improved nutrient utilization efficiency means that it increases photosynthetic rates [39]. Thus, plant growth has been studied using carbonated springs. In these studies, a quantitative study for the emitting of free carbon dioxide and $CO_2$ in carbonated spring water is needed. It is especially important to validate the experimental method without the influences of $CO_2$ supply from boilers, soils, and $CO_2$ from the atmosphere.

Carbonated springs exist mainly in volcanic areas where volcanic influences are thought to be. Hot spring waters originating from volcanos tend to become a low pH because many of them contain hydrogen chloride and hydrogen sulfide [40]. $CO_2$ dissolves in a water mainly in the form of free carbon dioxide when pH is low. Thus, hot spring water gushes to the surface with a high concentration of free carbon dioxide. It is easy to predict that the gushing will cause a sudden drop in atmospheric pressure, emitting $CO_2$ into the atmosphere. The presumed phenomenon as a carbonated spring with high salt concentration is that a large amount of free carbon dioxide is dissolved in the spring water. It is conceivable that free carbon dioxide dissolves more in the spring water with an even higher salt concentration due to being under pressure higher than atmospheric pressure. This suggests that a large amount of free carbon dioxide is emitted from the hot spring water as bubbles when the water gushes to the surface. This is exactly the same phenomenon when the lid of a plastic bottle of soda water is opened. The extent of the bubbles emitting from hot spring water may exceed the plastic bottles of soda water because the pressure is greater than the plastic bottle. Nagayu Spa as the research area of this study is also a carbonated spring with a high salt concentration (approximately 3546.5 mg/L). This hot spring water has a high concentration of free carbon dioxide due to the volcanic gases of Mt. Kuju, which is located near Nagayu Spa [41]. This environment is conducive to high levels of free carbon dioxide. The rapid pressure difference is created by the upwelling to the surface and it is expected to initiate a rapid $CO_2$ vaporization. However, Nagayu Spa does not produce many bubbles like carbonated water in appearance, and this hot spring water maintains its clear state for over ten minutes. Therefore, it is a low possibility for Nagayu Spa that free carbon dioxide is emitted due to the difference in partial pressure. It is unclear whether such hot spring water can be used for carbon dioxide to improve

agricultural production. The emitting of $CO_2$ should normally be visibly confirmed due to partial pressure; however, it is unclear whether $CO_2$ is really being emitted under conditions where bubbles are not visible. Even if $CO_2$ were emitted from hot spring water, it is unclear whether it would be possible to increase the $CO_2$ concentration in a greenhouse. In other words, previous studies are not clear regarding whether it is possible to increase the $CO_2$ concentration above 410 ppm. Furthermore, it is unclear whether volcanic hot spring water can be used to elevate $CO_2$. This needs to be clarified in a way that is not affected by boilers, soil, or atmosphere. If free carbon dioxide can increase $CO_2$ in a facility, it is necessary to clarify whether it can be used in actual horticultural agriculture. Therefore, it is necessary to determine what is responsible for the amount of free carbon dioxide emitted and to determine the scale of institutional horticulture that can use the free carbon dioxide contained in hot spring water. By solving these problems, how $CO_2$ in hot spring water can be used to promote plant growth can be determined. This will not require energy resources and will promote plant growth, thus contributing to solving food shortages in the future. This $CO_2$ will be released into the atmosphere if not utilized. If it is utilized, there is a possibility that the $CO_2$ concentration in the atmosphere can be reduced. Furthermore, hot spring water is a resource that can be used sustainably as long as it is not used in the wrong quantities, allowing for sustainable use and sustainable plant growth. In addition, the use of carbonated hot spring water is expected to have a positive impact on the economy as it reduces the use of energy resources and removes artificially created $CO_2$.

This study focused on free carbon dioxide contained in volcanic hot spring water and examined whether it is possible to increase the concentration of $CO_2$ in the greenhouse by using the incubation method. To clarify the free carbon dioxide emitting performance, the implementation time in the demonstration experiment, temperature and humidity in the incubator, and the amount of hot spring water were examined. The purpose of this study is to examine the possibility of applying the free carbon dioxide in hot spring water to greenhouse horticulture and plant factories. The first step is to confirm whether the target hot spring waters contain free carbon dioxide. Then, the relationship between the free carbon dioxide in the hot spring water and $CO_2$ emitted into the air will be clarified. In the process, this study will clarify the effects of experiment time, environmental conditions, such as temperature and humidity, and the amount of hot spring water on $CO_2$ emitted. The results will be used to examine how they can be applied to horticultural agriculture.

## 2. Materials and Methods

### 2.1. Outline of Target Hot Spring Water

The demonstration experiment was conducted using Hot Spa A and Hot Spa B in Nagayu Spa (Japanase name is Nagayu Onsen). Their hot spring waters are located in Taketa City, Oita Prefecture, in Japan, as shown in Figure 1. There are many hot spring waters in this area, and this area is called Nagayu Spa. Hot Spa A and Hot Spa B exist in Nagayu Spa; however, the sources are different. These hot spring waters were selected from among the many sources within Nagayu Hot Springs because a preliminary survey conducted by our laboratory showed high $CO_2$ concentrations within Nagayu Spa, and these could be freely collected in a survey. The preliminary survey was conducted to investigate the concentration of bicarbonate ions ($HCO_{3-}$). Characteristic of the Nagayu Spa is that it contains a high concentration of $CO_2$ and minerals. Therefore, it is expected to significantly increase $CO_2$ in a greenhouse, and it was determined to be the most suitable for the validation of this study. Such hot springs exist throughout the world, as indicated in Section 1. Therefore, the results of this study could be applied to those hot spring waters. The next section presents the experimental methodology used in this study for the hot spring water of Nagayu Spa.

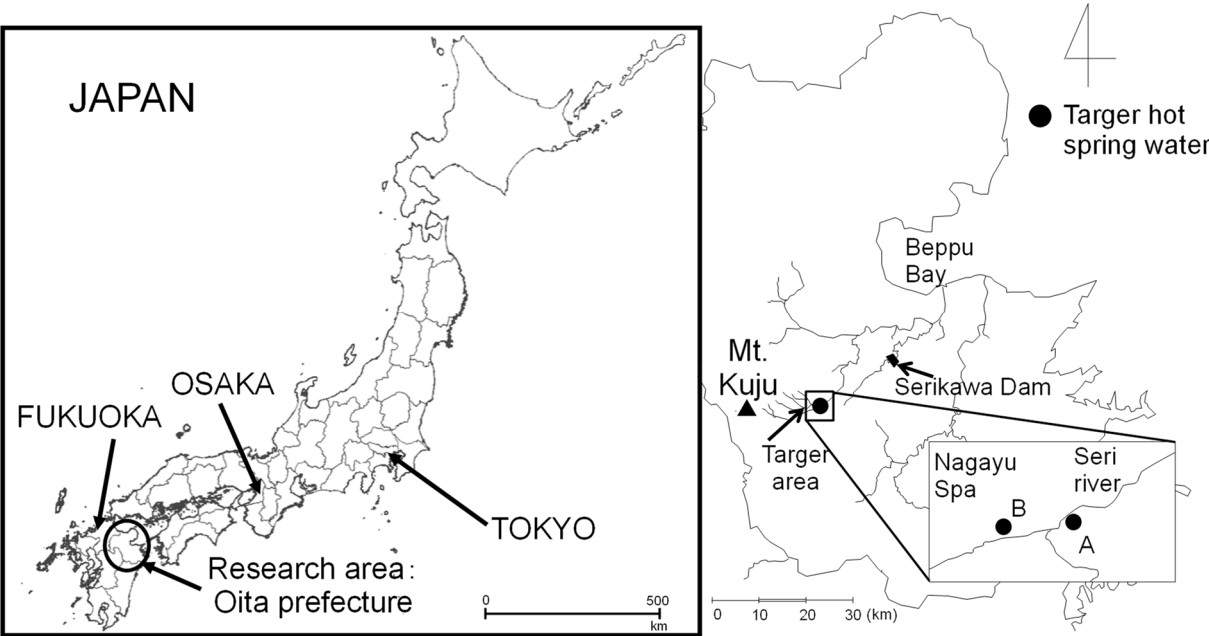

**Figure 1.** Study area and hot spring water (Hot Spa A and Hot Spa B). This map is based on the Digital Topographic Map published by Geospatial Information Authority of Japan of Ref. [42].

*2.2. Methods and Demonstration Experiment*

### 2.2.1. Water Sampling Method

Hot spring water transported by pipes from the source was collected directly into plastic sampling containers. The reason for using transported hot spring water was because it is difficult to extract these waters directly from the source. When collecting the specified volume of water (e.g., 1 L), a plastic measuring cylinder was used.

### 2.2.2. Free Carbon Dioxide Measuring Method

Free carbon dioxide was determined by titration with sodium hydroxide in accordance with the Japan Industrial Standard (JIS K 0102:2013, Japan Industrial Standard) [43]. This titration did not use the addition of the potassium oxalate because the pH of the hot spring water showed weak acidity. The titration was performed in the field using a burette, a stoppered flask, and a volumetric flask. There are several methods for measuring free carbon dioxide, and this titration method is the most commonly used method for on-site measurement. Other methods include analysis using a TOC meter performed in a laboratory. Free carbon dioxide has the characteristic of being released over time. Therefore, it is difficult to measure the original concentration with this method. A phenolphthalein solution was used as an indicator because hot spring waters were transparent and colorless. The titration measured the amount added of sodium hydroxide until a light red color was achieved. The titration was performed with 0.02 N sodium hydroxide and hand stirring. The timings of measuring for the free carbon dioxide contained in hot spring water were before the experiment and after the experiment. The concentrations of free carbon dioxide were calculated using the following Formula (1) from the amount of titration.

$$\text{The concentration of free carbon dioxide (mg/L)} = \text{normal (i.e., 0.02 normality)} \times \text{the amount of titration(mL)} \div \text{quantity of water (mL)} \times 62 \times 1000 \tag{1}$$

### 2.2.3. Measurements of pH, EC, ORP, Water Temperature, and Dissolve Components

pH (DKK-TOA Corp., Tokyo, Japan, model number: HM-30), EC (Electrical Conductance: HORIBA, Ltd., Kyoto, Japan, model number: D-24), and ORP (Oxidation-reduction potential: HORIBA, Ltd., Kyoto, Japan, model number: RM-20P) were measured by a

portable equipment. Water temperature was measured with an alcohol thermometer. Bicarbonate ion ($HCO_{3-}$) concentration was calculated by using the following Formula (2). The amount of titration was the amount of sulfuric acid (i.e., titration amount) when the pH value of sample became 4.8 by using 0.02 factor sulfuric acid. pH, EC, ORP, and water temperature were measured on site. For bicarbonate ion, samples were taken back to the laboratory for analysis.

$$\text{Bicarbonate ion concentration (mg/L)} = \text{normal (i.e., 0.02 normality)} \times \text{the amount of titration (mL)} \div \text{quantity of water (mL)} \times 61 \times 1000 \tag{2}$$

Dissolved components except bicarbonate ion were measured by the ion chromatography (Thermo Fisher Scientific, Waltham, MA, USA, model number: ICS-1000) in the laboratory after pretreatment using a 0.45 μm hydrophilic filter.

### 2.2.4. Method of the Emitting Experiment for Free Carbon Dioxide Contained in Hot Spring Water

For checking the free carbon dioxide emitting performance from hot spring water, the experiments were conducted to clarify the effects of implementation time in a demonstration experiment, temperature and humidity in an incubator, and amount of hot spring water. The reason for using an incubator is to conduct the experiment under conditions that minimize the influence of $CO_2$ from the outside air. Plastic greenhouses and other facilities would not be able to completely block the entry of outside air and could also be affected by soil and other factors. An incubator is also equipped with a fan, which can be used to generate air circulation. The general method of experiments using incubation involves constant temperature through the operation of heaters and air circulation through the operation of fans. In this study, in order to confirm the amount of free carbon dioxide emitted from hot spring waters under natural conditions, the experiment was conducted with only fans running and no heaters [44]. The months of the experiment were in August, September, October, and December 2021 and March 2022.

The experiment used the incubator, as shown in Figure 2. According to the manual, the performance of the equipment was as follows. The shape and size of the incubator were a square prism with effective internal dimensions of 305 mm × 285 mm × 250 mm. The volume was $2.17 \times 10^7$ mm$^3$. The small size reduces the difference of $CO_2$ concentration in the incubator and is a good way to check the effects of the emitting from hot spring water. One of the problems in managing $CO_2$ in horticultural facilities is that the large volume of the facility and inadequate circulation within the facility can cause differences in $CO_2$ concentration within the facility. This causes differences in plant growth. The small size of the system allows for sufficient circulation to accurately measure the amount emitted from hot spring water. Assuming application for actual institutional horticulture, the size of this incubator is a very small volume. Since the concentration of $CO_2$ is expressed per volume (ppm), its application in horticultural facilities can be determined by calculation if the difference in volume of the facilities is quantitatively clear. The size of the incubator used in the experiment has no significant effect on its application to horticultural facilities. The incubator used in this study can minimize air exchange with the outside. Therefore, the amount of free carbon dioxide emitted from the hot spring water and the amount of $CO_2$ increase in the incubator can be studied for various conditions such as the amount of hot spring water.

The incubator was equipped with a fan and a heater. The capacity of the fan was 0.32 m$^3$/min. One of the sides had a hinged door with stopper for closing the door with a handle. The inside of the incubator was divided into three layers (vertical length: 75, 195, 115 mm) with a perforated partition plate. The air in the incubator could be circulated. Hot spring water in a container and a stirrer for its base purpose were put in the middle layer. $CO_2$ data loggers were installed in the upper and the lower layers. This study presents results of the $CO_2$ data logger as A shown in Figure 2, which was installed in the upper

layer directly affected by the hot spring water. The interval of data acquisition by the data logger was every 10 s.

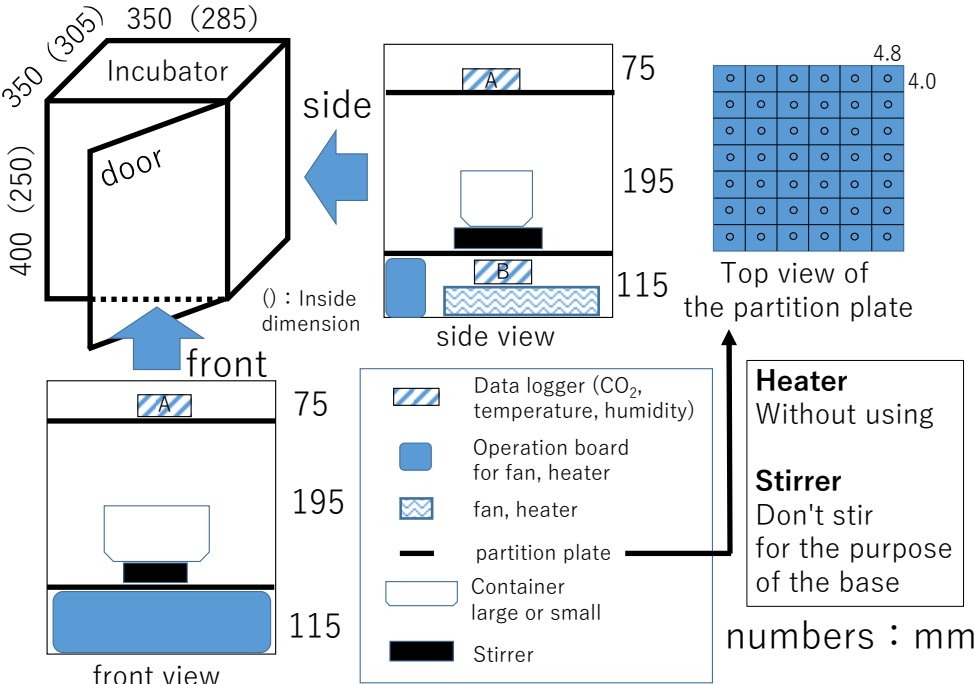

**Figure 2.** Schematic diagram for the used incubator.

Experimental conditions were the amount of hot spring water (i.e., 500, 1000, and 2000 mL), closing the door, and running or stopping the fan. The size of the containers containing the hot spring water were 159 mm × 124 mm × 80 mm for 500 mL and 1000 mL, 276 mm × 152 mm × 93 mm for 2000 mL. The time of the demonstration experiment meant from the start of the experiment to the end of the experiment. The reason for conducting the experiment under closed-door conditions was to prevent the air in the incubator from exchanging with the outside air. However, $CO_2$ concentration, temperature, and humidity in the incubator replaced the outside air when the next experiment was performed in the same incubator. The heater was not in operation because it is assumed to be applied to actual greenhouse horticulture.

## 3. Results and Discussion

This chapter first presents basic water quality results, such as pH, EC, ORP, and concentrations of dissolved constituents in hot spring water. For grasping the performance of the emitting of free carbon dioxide contained in hot spring water, it determined the fluctuation of free carbon dioxide concentration in hot spring water and $CO_2$ concentration in the incubator. Furthermore, it clarified the effects of the implementation time in the demonstration experiment, temperature and humidity in the incubator, and amount of hot spring water for emitting free carbon dioxide. The intent of examining the relationship between the implementation times and emitting of free carbon dioxide was to make a unified decision because the time of each experiment was different. In this experiment, the temperature did not keep at a constant level. The experiment was conducted without a heater to bring the conditions closer because the actual horticultural farming is affected by an ambient air temperature. Although it is necessary to examine each factor comprehensively using regression analysis for what needs to be resolved as in the references, this study first examined each of the targeted factors one by one to check their effects [45].

### 3.1. Water Quality of the Hot Spring Water

Figures 3 and 4 show the results of annual fluctuation of pH and water temperature for Hot Spa A and Hot Spa B.

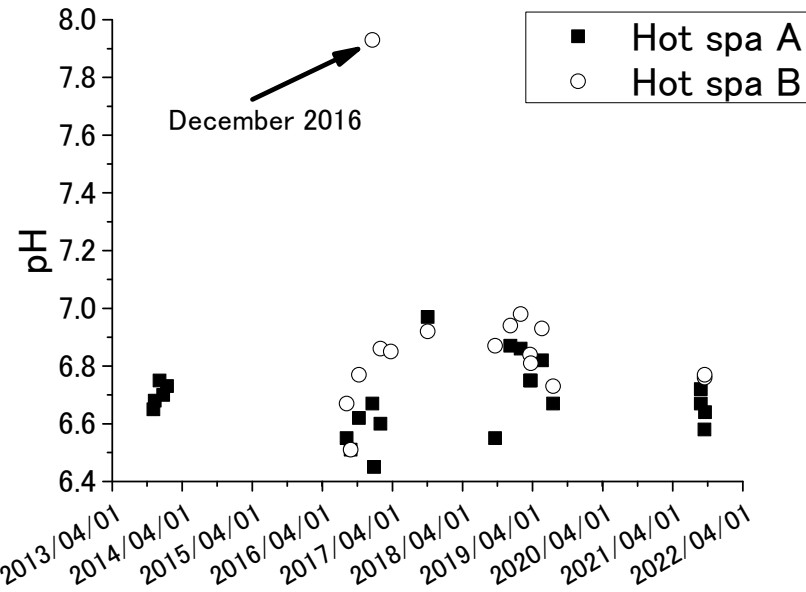

**Figure 3.** Annual fluctuations of pH for Hot Spa A and Hot Spa B.

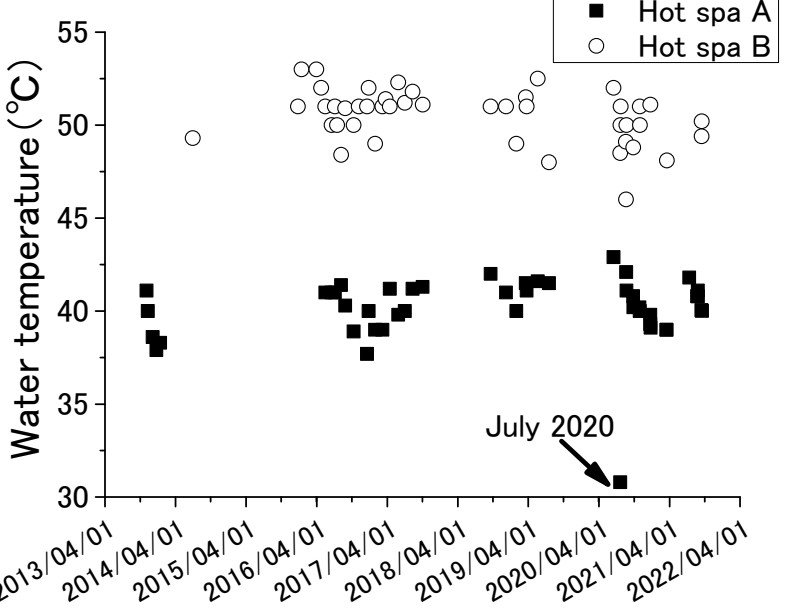

**Figure 4.** Annual fluctuations of water temperature for Hot Spa A and Hot Spa B.

Tables 1 and 2 show the results of average values and standard deviations of pH, EC, ORP, water temperature, carbonate ion, and the total amount of dissolved components, including bicarbonate ion (i.e., $Na^+$, $Mg^{2+}$, $Ca^{2+}$, $Cl^-$, $SO_4^{2-}$, $HCO_{3-}$) for Hot Spa A and Hot Spa B. Those were measured by using the methods presented in Section 2.2.3. pH, EC, ORP, water temperature, and $HCO_{3-}$ were measured in the field, while other dissolved constituents were analyzed by ion chromatography in the laboratory.

**Table 1.** Average and standard deviation for various water quality (Hot Spa A).

| Hot Spa A | pH | EC (mS/m) | ORP (mV) | Water Temperature (°C) | HCO$_3-$ (mg/L) | Dissolved Components (mg/L) |
|---|---|---|---|---|---|---|
| Average | 6.7 | 489 | −58 | 40.2 | 2184.0 | 3546.5 |
| Standard deviation | 0.1 | 24 | 8 | 1.8 | 95.5 | 288.3 |
| Number of samples | 23 | 17 | 18 | 47 | 32 | 30 |
| | | | | | | HCO$_3-$ only 32 |

Dissolved components: Na$^+$, K$^+$, Mg$^{2+}$, Ca$^{2+}$, Cl$^-$, SO$_4^{2-}$, HCO$_3-$. The standard deviation of dissolved components was the sum of all the standard deviations of each ion.

**Table 2.** Average and standard deviation for various water quality (Hot Spa B).

| Hot Spa B | pH | EC (mS/m) | ORP (mV) | Water Temperature (°C) | HCO$_3-$ (mg/L) | Dissolved Components (mg/L) |
|---|---|---|---|---|---|---|
| Average | 6.9 | 646 | −76 | 50.3 | 2859.3 | 4682.4 |
| Standard deviation | 0.3 | 120 | 25 | 1.4 | 227.8 | 465.2 |
| Number of samples | 16 | 13 | 13 | 46 | 47 | 49 |
| | | | | | | HCO$_3-$ only 47 |

Dissolved components: Na$^+$, K$^+$, Mg$^{2+}$, Ca$^{2+}$, Cl$^-$, SO$_4^{2-}$, HCO$_3-$. The standard deviation of dissolved components was the sum of all the standard deviations of each ion.

The results of ORP were the measured data, not corrected to the hydrogen electrode potential. The investigation period was November 2013 to September 2021 for pH and November 2013 and July 2019 for the other chemical components. Sample number is shown in the table. The averages and standard deviations of pH and water temperature in the tables were calculated from the results of Figures 3 and 4. pH showed between 6.4 and 7.0, except 7.9 for Hot Spa B in December 2016. The pH of Hot Spa A in the same month was 6.7. It was difficult to determine the cause. Forms of carbon dioxide contained in hot spring water are generally carbonate ion (CO$_3^{2-}$), bicarbonate ion (HCO$_3-$), and free carbon dioxide (H$_2$CO$_3$ or CO$_2$ (aq)). The proportion of each carbon dioxide can be calculated by pH, except a strongly acidic hot spring water. The proportion of free carbon dioxide is highest when the pH value is between 4 and 6.5 [46]. The proportion of bicarbonate ions increases when the pH value exceeds 6.5. Free carbon dioxide is completely changed to carbonate and bicarbonate ions when the pH value exceeds 8.3. The average of pH values of Hot Spas A and B were 6.7 and 6.9. This pH showed a high proportion of bicarbonate ions and free carbon dioxide, although it was outside of the highest value for the percentage of free carbon dioxide. Water temperature for Hot Spa A was around 40 degrees, and Hot Spa B was around 50 °C. Water temperatures tended to be higher during periods of high air temperatures. Conversely, water temperatures were high when air temperatures were high. The water temperature of Hot Spa A was 30.8 °C in July 2020, which was about 10 °C lower in annual fluctuations. Hot Spa A was inundated by flooding the of the Seri River near Hot Spa A before the survey in July 2020. It presumes that this was a factor in the temporarily low water temperature. ORP showed a negative value, indicating that the subject hot spring water is in a reduced state. Since much of a subsurface is a reducing environment with poor oxygen, it is possible that it has existed underground for an unknown period of time and has been affected in some way underground. Based on the above, since the target hot spring water contains free carbon dioxide, it is possible to verify the increase in atmospheric CO$_2$ due to free carbon dioxide. In addition, the water temperature is as high as 40 °C or 50 °C, which means that the groundwater can be utilized as a resource.

### 3.2. Relation between Free Carbon Dioxide and Carbon Dioxide

Figures 5–8 show concentration fluctuations for free carbon dioxide in hot spring water and $CO_2$ in the incubator. The horizontal axis shows the time of the experiment and the vertical axis shows the $CO_2$ change in the incubator as measured by the $CO_2$ data logger. The experiment in Figure 5 was conducted for the longest time, more than 4 h, while the rest of the experiments in Figures 6–8 were completed within 2 h. For Figures 6–8, the experiment was terminated because the concentration of free carbon dioxide in the hot spring water had decreased. The amounts of hot spring water were 1000 mL, 2000 mL, and 3000 mL using a 1000 mL vessel (i.e., 159 mm × 124 mm × 80 mm) and 2000 mL vessel (i.e., 276 mm × 152 mm × 93 mm; it can hold over 3000 mL). The time intervals for measuring free carbon dioxide were indeterminate. The down arrows in the figure indicate the timing for the measurement of free carbon dioxide, hence the caption "Timing to take out the sample" as taking the time the sample was entered for these figures. The operation method of the fan was different depending on the experimental conditions, as shown in the figure. The phenol check in Figure 8 meant that the concentration of free carbon dioxide was not 0 mg/L, but it was a very small concentration. If the sample color turns red when a phenolphthalein solution was added to the sample, it means that it does not contain free carbon dioxide (JIS K 0102:2013, Japan Industrial standard) [43]. The red color of the end point was not observed when the phenolphthalein was added to the sample for measuring free carbon dioxide. The sample turned red color when a small quantity of sodium hydroxide was added, but it was difficult to measure the titration volume with a burette. Therefore, these samples were designated as a phenol check in this paper. To measure free carbon dioxide contained in the hot spring water, the door was opened to take the container with hot spring water from the incubator. For the analysis of free carbonic acid, the amount of hot spring water necessary for the measurement was taken from the container. The container was then returned to the incubator and the door was completely closed. This incubator exchanged the inside air and outside air less when its door was closed. The air in the incubator and the outside air were exchanged while opening the door.

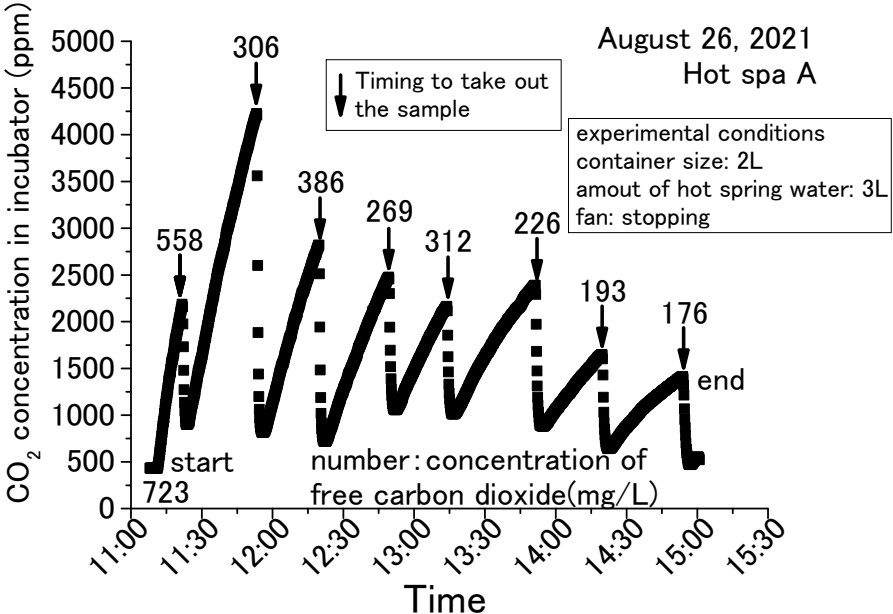

**Figure 5.** Concentration change of free carbon dioxide contained in hot spring water for 3 L obtained with a 2 L container (Hot Spa A on August 2021).

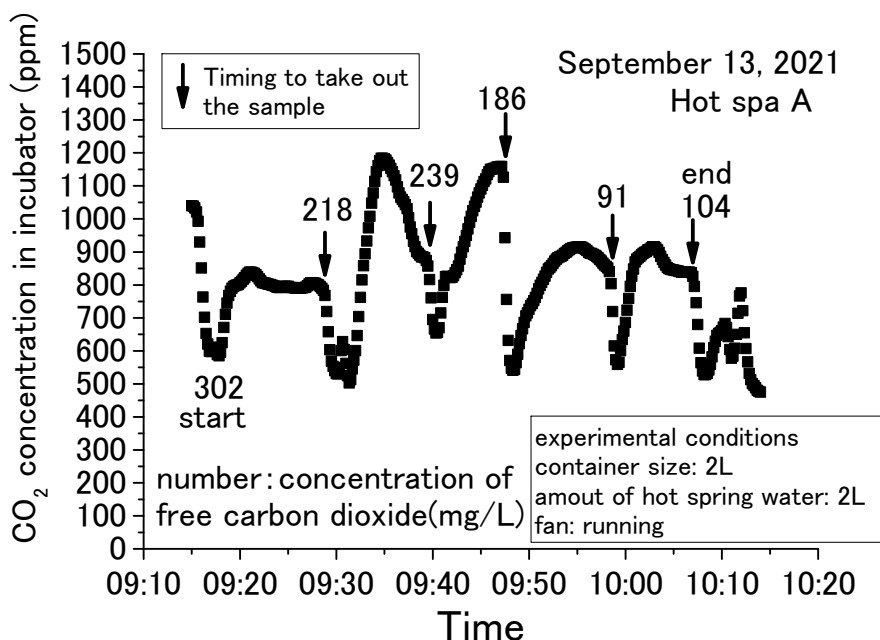

**Figure 6.** Concentration change of free carbon dioxide contained in hot spring water for 2 L obtained with a 2 L container (Hot Spa A on September 2021, part 1).

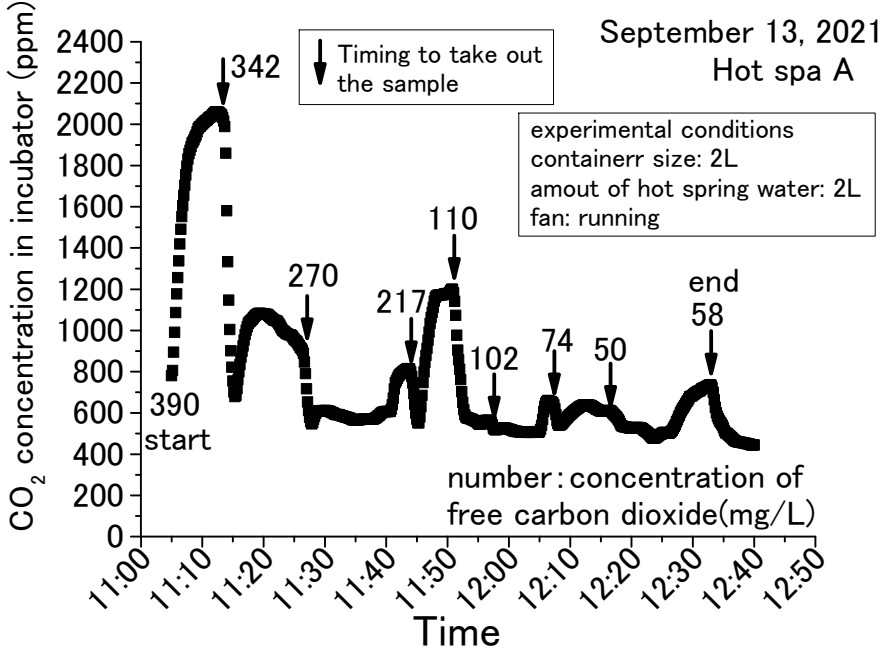

**Figure 7.** Concentration change of free carbon dioxide contained in hot spring water for 2 L obtained with a2 L container (Hot Spa A on September 2021, part 2).

The common trend in Figures 5–8 is that the $CO_2$ in the incubator decreased when the sample was taken out of the incubator. It could be assumed that the concentration decreased by opening the door and exchanging the air inside the incubator with the outside air. The decreased $CO_2$ concentration increased when the hot spring water was returned to the incubator and the door was closed. This trend could be confirmed over and over again. In this demonstration experiment, there was no supply of $CO_2$ from sources other than the hot spring water.

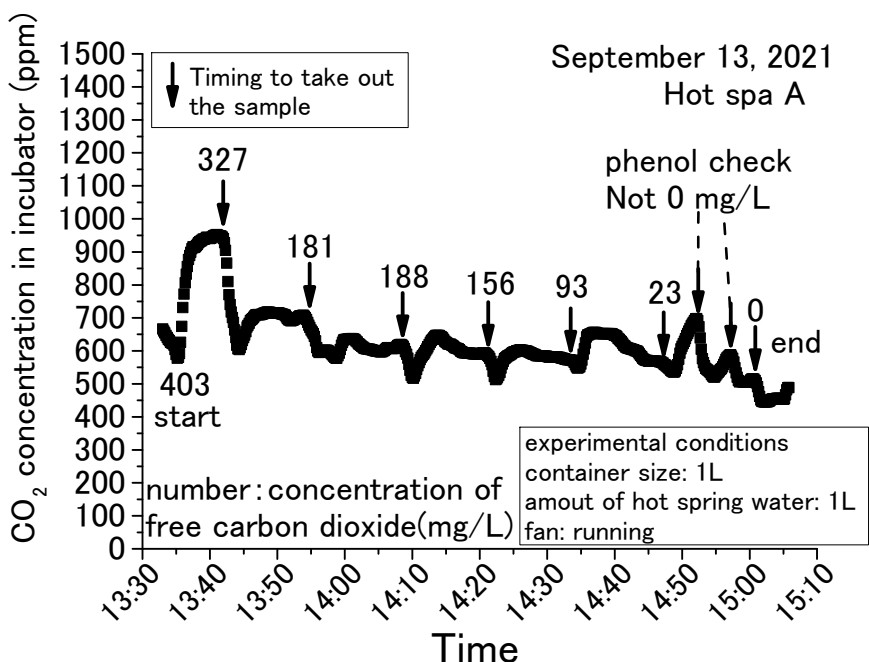

**Figure 8.** Concentration change of free carbon dioxide contained in hot spring water for 1 L obtained with a 1 L container (Hot Spa A on September 2021 part 3).

Checking the results, the concentration of $CO_2$ increased over time, while the concentration of free carbon dioxide decreased. $CO_2$ concentration in the incubator increased by emitting the free carbon dioxide contained in the hot spring water. The concentration of free carbon dioxide tended to decrease with the passage of experimental time. When the amount of hot spring water was 2 or 3 L, free carbon dioxide remained in the hot spring water even when the experiment was terminated. When the amount of hot spring water was 1 L, it showed a phenol check and the concentration was very low. In addition, for 1 L, the amount of $CO_2$ concentration in the incubator increased by a small amount. This means that $CO_2$ emitting will continue until the amount of free carbon dioxide is very low. $CO_2$ changes were significantly different in the incubator whether the fan was running or stopped. $CO_2$ concentration showed an almost constant rise in the case of stopping the fan, as shown in Figure 5. On the other hand, the change of concentration showed slowly or decreased after a certain concentration in the case of running the fan, as shown in Figures 6–8. When the fan was stopped, the maximum value of $CO_2$ concentration in the incubator was about 4000 ppm as shown in Figure 5. When the fan was running, the maximum value of $CO_2$ concentration in the incubator was about 2000 ppm, as shown in Figure 7. The maximum value without the fan greatly exceeded the maximum value with the fan attached to the incubator. Therefore, the maximum $CO_2$ values tended to be larger without fan operation. This factor can be assumed to be due to the air in the incubator being circulated and stirred by the fan, eliminating the shading of $CO_2$ concentration. This study shows results from the $CO_2$ data logger placed directly above the hot spring water. Therefore, it is possible that $CO_2$ collected directly above the hot spring water and increased its concentration. As mentioned in the introduction, it can be inferred from previous studies that without the use of fans to circulate air in a greenhouse, there will be shading in the concentration of $CO_2$. Figures 6 and 8 were differed with the amount of hot spring water. Those results showed that the $CO_2$ concentration in the incubator was higher when the amount of hot spring water was larger. The concentration of free carbon dioxide in the hot spring water became 0 mg/L more quickly when the amount of hot spring water was smaller, as is shown in Figures 7 and 8. This was thought to be due to the fact that when the amount of hot spring water was low, the amount of free carbon dioxide contained was also low. Therefore, it was found that a larger volume of hot spring water

was able to increase the $CO_2$ concentration in the incubator and continue to emit the free carbon dioxide for a longer time. Thus, depending on the amount of water, there is very little free carbon dioxide remaining in the hot spring water. Nevertheless, within 60 to 70 min after the start of the experiment, free carbon dioxide remained in the hot spring water. The maximum value of $CO_2$ concentration was shown at an experimental time in Figures 5–8. Figures 6 and 7 were the same conditions regarding the amount of hot spring water, the size of container, and the operation of the fan. The experiment times were about 25 min for Figure 6 and 10 min for Figure 7 when the $CO_2$ concentration was shown to be the maximum value. This study was unable to confirm the relationship between the maximum of $CO_2$ concentration and the experimental time. From the above, it was found that the concentration of $CO_2$ in the greenhouse can be increased by using free carbon dioxide contained in hot spring water. Free carbon dioxide remained in the hot spring water even after 60–70 min had passed since the start of the experiment. It was also found that there were differences depending on the amount of hot spring water and whether or not fans were in operation. Therefore, all subsequent results are shown with the fan running. However, the experimental time did not show the relationship with the $CO_2$ concentration in the incubator for these experiments.

### 3.3. The Effect of the Implementation Time in Demonstration Experiment

With each passing hour of the experiment, the free carbon dioxide in the hot spring water decreased and the $CO_2$ concentration in the incubator increased. Some relationship was shown for the running or stopping of fan operation and the amount of hot spring water. It has not yet been possible to determine a relationship by time of implementation. For checking the change in free carbon dioxide in hot spring water with time in the experiment, it is shown in Figures 9 and 10 that the implementation time in the demonstration experiment and concentration difference of free carbon dioxide contained in hot spring water as Hot Spa A and Hot Spa B before and after the experiment. In addition, for checking the $CO_2$ changes in the incubator over the time of the experiment, the implementation time in the demonstration experiment and the concentration difference of $CO_2$ in the incubator before and after the experiment when Hot Spa A and Hot Spa B are set in the incubator are shown in Figures 11 and 12. The data covered are those with an experimental duration of 70 min or less in which free carbon dioxide remained. Hot Spa A was selected for less than 70 min, and Hot Spa B was selected for less than 50 min. Also, for checking the impact of different water volumes, these figures were shown separately for each amount as 500 mL, 1000 mL, and 1500 mL of hot spring water. In the caption, it shows the values for each amount of water and the equations and coefficients of determination for each amount of water. The fan is running as an experimental condition. Experiments were conducted at different implementation times for confirming the trend of concentration difference by time of implementation. The slope of the correlation equation showed a negative value, except 1000 mL in Figure 9, 1000 mL in Figure 11, 500 mL in Figure 12, and 1000 mL in Figure 12. The negative values, shown Figures 9 and 10, meant that the amount of emitted free carbon dioxide from hot spring water was higher at the beginning of the experiment. The reason for the greater emission at the beginning of the experiment was that the partial pressure of the hot spring water was higher, making it easier for $CO_2$ to be emitted. It could be assumed that as time passes, the partial pressure difference with the air in the incubator became smaller and less likely to be emitted. The efficiency for emitting of free carbon dioxide was found to be low even after the experimental time was becoming longer. The negative values shown in Figures 11 and 12 meant that shortening the time of the experiment increased the increment quantity of $CO_2$ concentration in the incubator. A shorter experimental time was more efficient for the emitting of free carbon dioxide from the hot spring water and for the increase in $CO_2$ concentration in the incubator. The coefficient of determination was 0.50 or higher on the numbers. There were few cases that showed a clear tendency such as a proportional relationship because there were some results in which the execution time was different even with the same concentration difference. Those results were the same

change even if it checked the difference in the amount of hot spring water. Therefore, the implementation time in the demonstration experiment had no influence on the change of the concentration for free carbon dioxide and $CO_2$ concentration in the incubator. Although the slope of the equation was positive and the coefficient of determination was low, it was possible that a short period of time was ideal for the efficient emitting of free carbon dioxide from hot spring water and for the efficient rise of $CO_2$ concentration in the incubator. In other words, replacing the hot spring water in a short period of time may be more efficient.

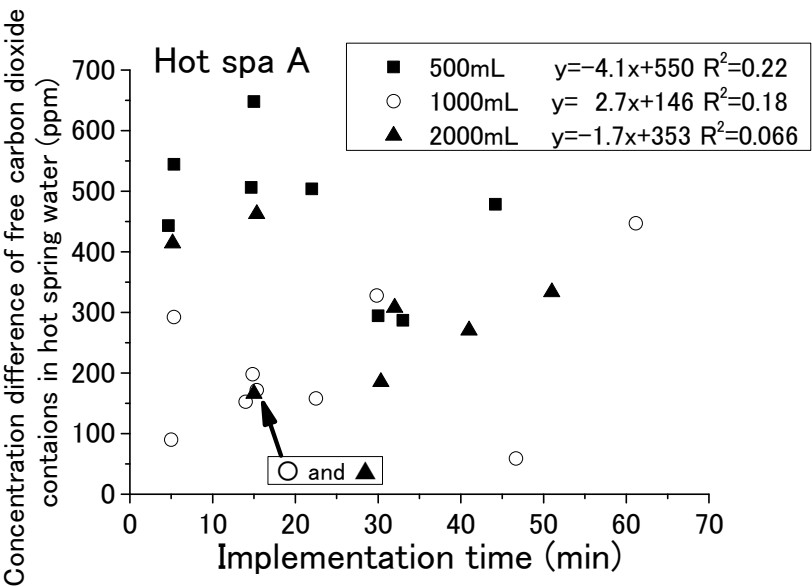

**Figure 9.** Relationship between implementation time and free carbon dioxide (Hot Spa A).

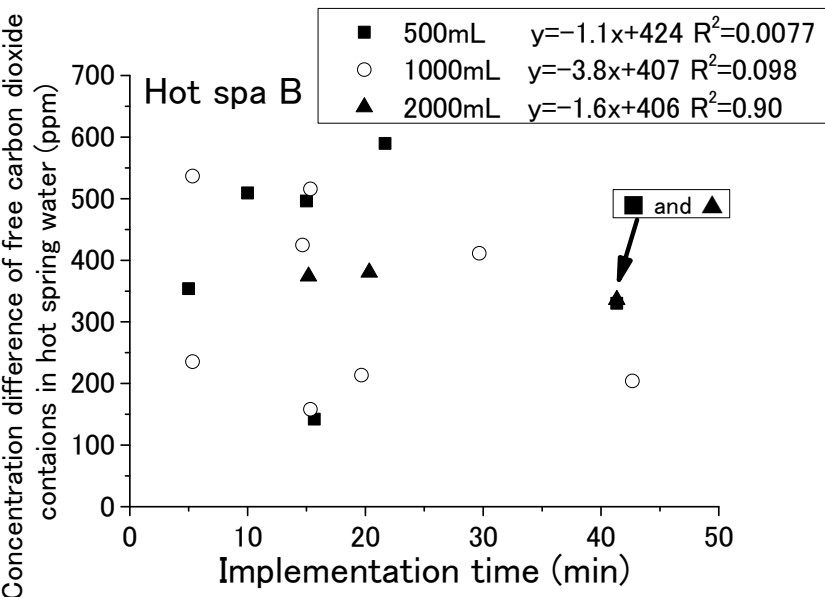

**Figure 10.** Relationship between implementation time and free carbon dioxide (Hot Spa B).

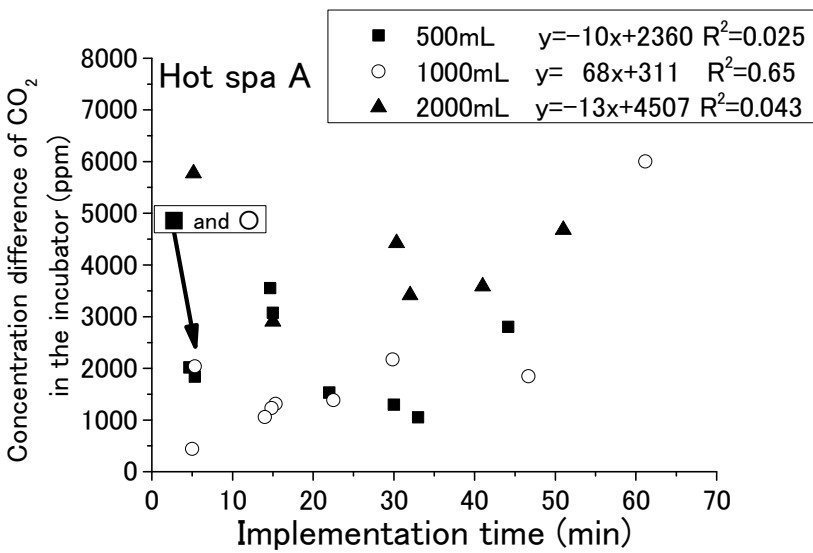

**Figure 11.** Relationship between implementation time and concentration difference of $CO_2$ in the incubator (Hot Spa A).

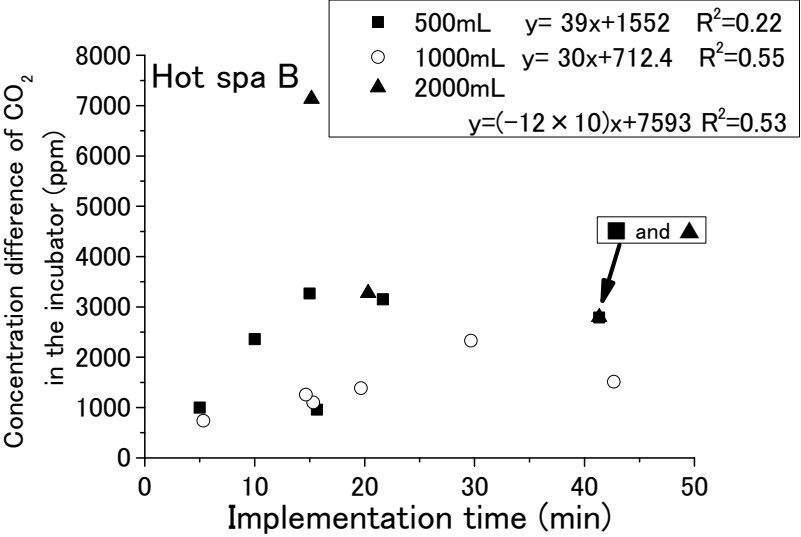

**Figure 12.** Relationship between implementation time and concentration difference of $CO_2$ in the incubator (Hot Spa B).

### 3.4. The Effects of Temperature and Humidity in the Incubator

Temperature and humidity in the facility may affect the amount of free carbon dioxide emitted. These values vary depending on the season and the surrounding environment. These changes may result in a decrease or no emitting of free carbon dioxide contained in the hot spring water. Therefore, it is necessary to check whether the amount of free carbon dioxide in the hot spring water and the $CO_2$ concentration in the incubator change with temperature and humidity. Samples must be taken out of the incubator for measuring free carbon dioxide. This action will change the temperature and humidity in the incubator. Therefore, it is difficult to ascertain the effects of temperature and humidity on the concentration of free carbon dioxide in hot spring water using the incubator method. In this paper, only changes in $CO_2$ concentration in the incubator will be checked to determine the relationship between temperature and humidity. For each result, the data cannot be checked under the same conditions due to differences in experimental time. As a way to confirm the results under the same conditions, the increase in $CO_2$ concentration per minute was calculated by dividing the increase in $CO_2$ in the incubator by the experimental time (minutes).

Figures 13–16 show that $CO_2$ concentration converted per minute at the beginning and the end for temperature and humidity in incubator. $CO_2$ concentration converted per minute means the value of concentration difference before and after the experiment divided by the experiment time. For each result, the experimental time was different. The amount of $CO_2$ increase per minute is shown for removing the influence of differences in the experimental time. As for the air temperature and humidity, they are the air temperature and humidity in the incubator. Temperature and humidity are increased by placing the hot spring water in the incubator. The experiment is conducted with the fan running and the door closed. Regarding the caption "Start of experiment", it indicates temperature or humidity in the incubator before the experiment as just before placing the hot spring water, and "End of experiment" indicates the temperature or humidity in the incubator after the experiment. Thus, there are two results for one change in $CO_2$ concentration per minute. For example, Figure 13 shows two results in the area of an approximately 1100 ppm $CO_2$ concentration, one for the temperature at the start of experiment and the other for the temperature after the end of the experiment. After the experiment, the results are concentrated on the right side of the figure due to higher temperatures and humidity. Although it would be sufficient to show only the results after the experiment, the results before the experiment are also shown.

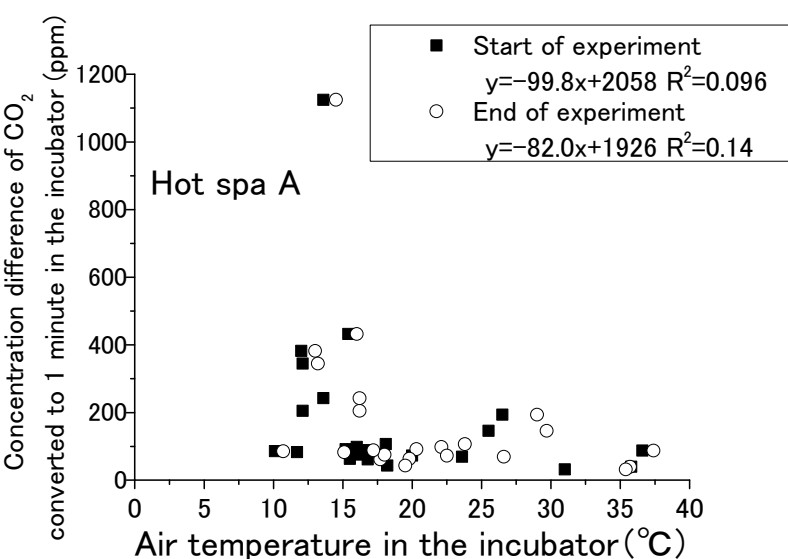

**Figure 13.** Relationship between air temperature in the incubator and concentration difference of $CO_2$ converted in 1 min in the incubator (Hot Spa A).

The water temperatures of the hot spring waters were 40 °C for Hot Spring A and 50 °C for Hot Spring B, which were always higher than the air temperature. The lower the temperature was, the greater the heat release from the hot spring was. Free carbon dioxide was also released accordingly, and the increase in $CO_2$ in the incubator was expected to be greater. The correlation formula shows a negative trend in Figure 13. The concentration difference was statistically smaller in the case of the higher temperature. It could not confirm a distinct shift for the amount of $CO_2$ increase when the air temperature was lower, such as by 10 to 15 °C, because it showed both high and low concentrations at the same air temperature. There was no particular relationship when the temperature exceeded 20 °C because the concentration difference was between 0 and 200 ppm. The trending of $CO_2$ increases for the changing of the air temperature was lacking exactness due to the low coefficient of determination at 0.096. The correlation of Hot Spa B on Figure 14 shows a positive slope; however, the coefficient of determination was low at 0.10 or less. Hot Spa B was not shown to clearly affect the air temperature in the incubator either. In general, it is easy to emit water vapor from hot spring waters with lower atmospheric humidity. The emitting performance of free carbon dioxide is predicted to change depending on the

amount of water vapor released into the atmosphere due to differences in atmospheric humidity. On Figure 15, the increment of $CO_2$ tended to decrease as the humidity increased for the expected result. In other words, it showed the same trend as the temperature results: the correlation coefficient was negative. The statistical relationship was small because the coefficient of determination was low. The same trend was observed for Hot Spa B in Figure 16. However, this expected result cannot be proven certain due to the low coefficient of determination. The effect of emitting $CO_2$ quantity on humidity was small for both Hot Spa A and Hot Spa B. The reason for the lower correlation coefficient is that the amount of the $CO_2$ increment was small even in low humidity. On the contrary, the amount of the $CO_2$ increment was large when the humidity was high. It was found that the difference in humidity did not affect the amount of the $CO_2$ increase. It could be inferred that the differences in air temperature and humidity in the incubator have not affected the amount of the discharge for free carbon dioxide.

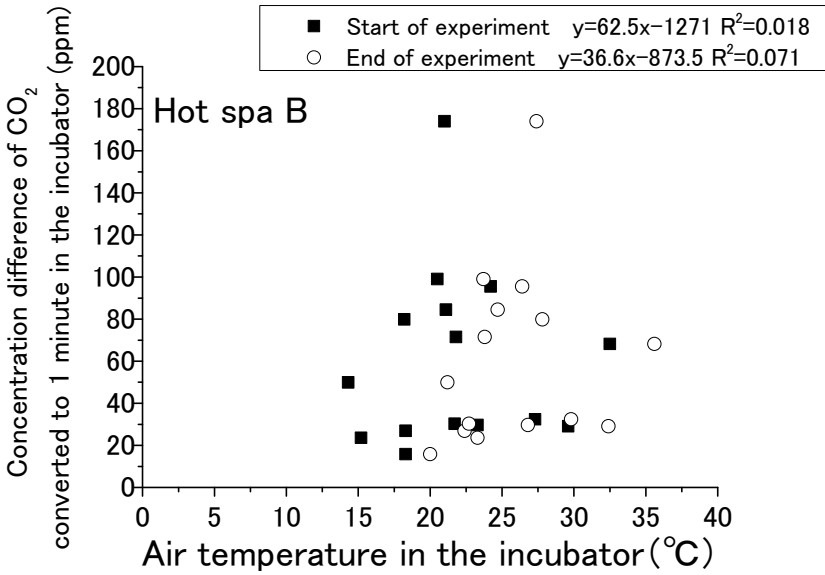

**Figure 14.** Relationship between air temperature in the incubator and concentration difference of $CO_2$ converted in 1 min in the incubator (Hot Spa B).

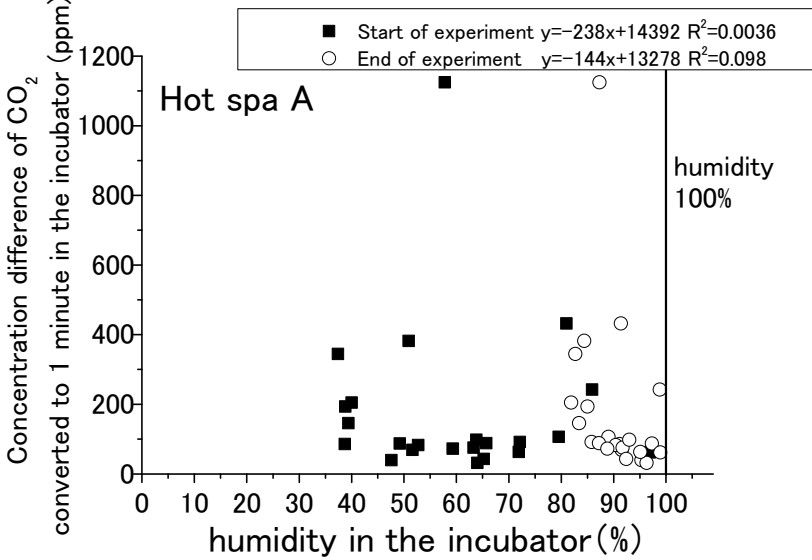

**Figure 15.** Relationship between humidity in the incubator and concentration difference of $CO_2$ converted in 1 min in the incubator (Hot Spa A).

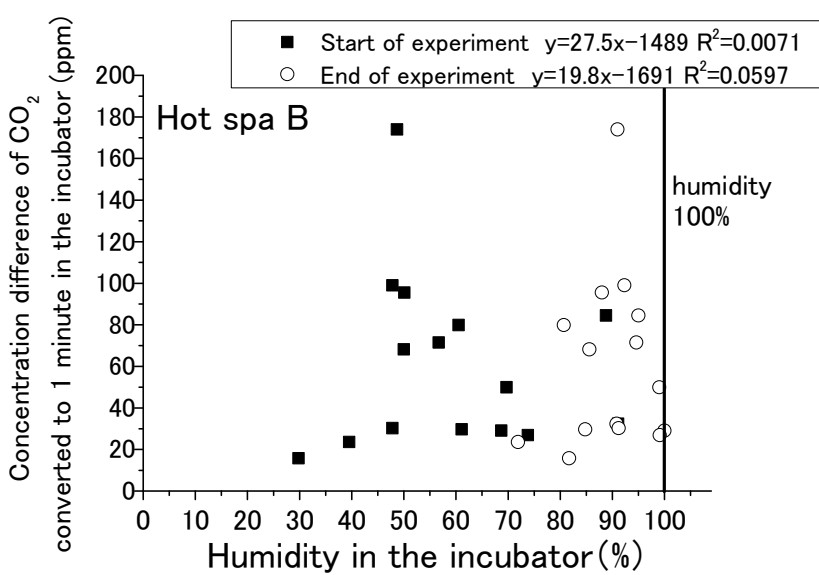

**Figure 16.** Relationship between humidity in the incubator and concentration difference of $CO_2$ converted in 1 min in the incubator (Hot Spa B).

### 3.5. The Effect of the Water Quantity of the Hot Spring

Finally, this study will check the effect of the amount of hot spring water on the $CO_2$ concentration in the incubator. To check the effect of the amount of hot spring water, the water volumes were set to 500 mL, 1000 mL, and 2000 mL. Therefore, Figures 17 and 18 show the relation between the concentration differences of free carbon dioxide contained in hot spring water and the $CO_2$ concentration difference in the atmosphere of the incubator. The concentration difference meant a difference between the concentration before the experiment and the concentration after the experiment. Therefore, the difference in the concentration of the free carbon dioxide in the hot spring water is the concentration of free carbon dioxide measured at the beginning of the experiment minus the concentration of free carbon dioxide measured in the container of the hot spring water taken out from the incubator after the emitting experiment in the incubator was completed (e.g., if the concentration at experiment starts is 500 mg/L and ends is 100 mg/L, the value subtracted is 400 mg/l.). The difference in $CO_2$ concentration in the incubator is also calculated by subtracting the value measured by the $CO_2$ data logger at the beginning of the experiment from the value when the spa water is taken after the emitting experiment in the incubator is completed (for example, if 400 ppm was measured at the start of the experiment and 1000 ppm at the end, the value subtracted would be 600 ppm.).

The results are shown for Hot Spa A and Hot Spa B, divided into 500, 1000, and 2000 mL of hot spring water. The coefficient of determination for Hot Spa A found a slight correlation that was over 0.47 (i.e., 0.47, 0.51, and 0.67). The slope showed 5.09 and 10.7 in the results of 500 mL and 1000 mL. However, it showed 5.09 and 6.76 as the numerical value of an approximation except the concentration differences 1846 and 6000 ppm. Relationships of 500 mL and 1000 mL showed a continuous trend bordering on 300 ppm for the concentration of free carbon dioxide. The intercept of 2000 mL was markedly different from 500 mL and 1000 mL. For Hot Spa B, the coefficient of determination for 500 mL showed the relationship. However, 1000 mL and 2000 mL showed almost no correlation (i.e., the coefficients of determination were 0.0016 and 0.22). Part of the 2000 mL sample showed results of a more than 7000 ppm difference for $CO_2$ concentration. It was well deviated from other trends. Hot Spa B did not show clear differences in trends compared with Hot Spa A. Based on the above, the coefficient of determination results showed a relationship for the same amount of water. For the different amounts of water, the slope was almost the same for Hot Spa A, except for the significant outlier values. However, the intercepts

yielded different results. Therefore, it was a possible that the amount of water has some effect on the intercepts. Therefore, it was necessary to consider what trends would be observed if converted to the same amount of water. In addition to the amount of hot spring water, the concentration of free carbon dioxide may have an effect on the results, so it was necessary to consider the trend per 1 ppm of free carbon dioxide.

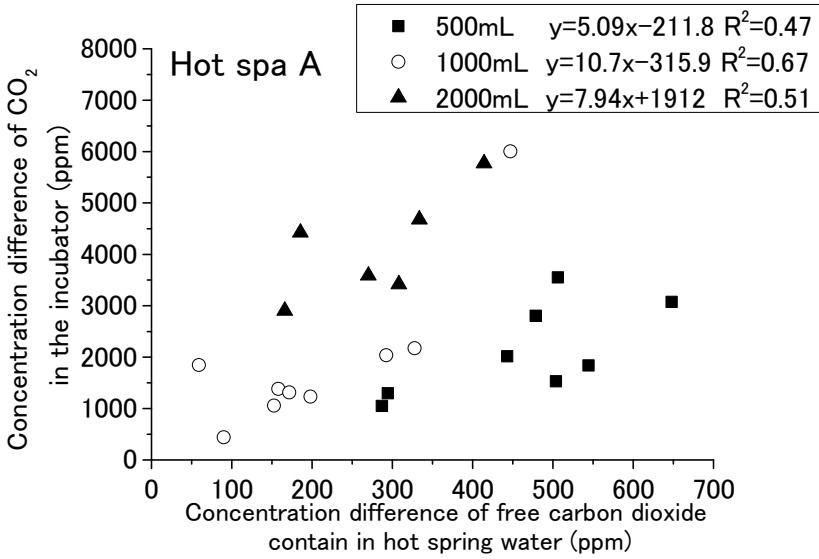

**Figure 17.** Relationship between concentration difference of free carbon dioxide contained in hot spring water and concentration difference of $CO_2$ in the incubator (Hot Spa A).

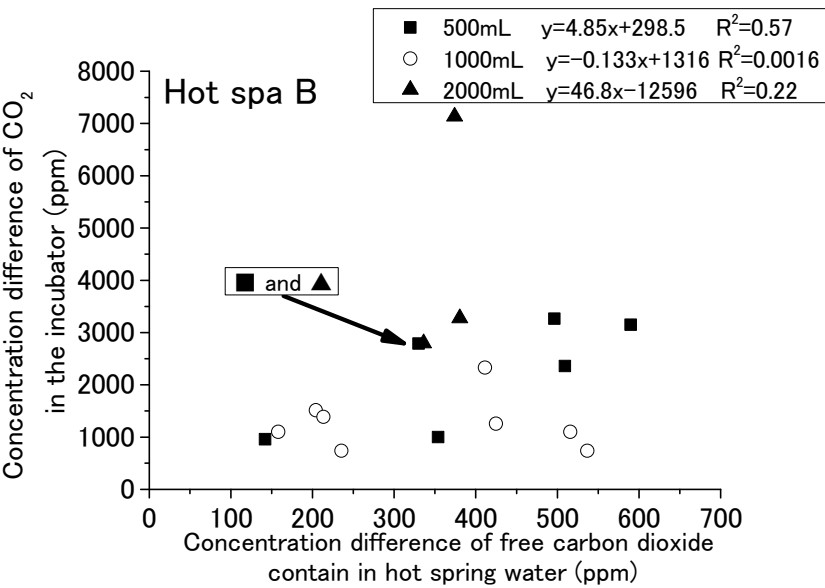

**Figure 18.** Relationship between concentration difference of free carbon dioxide contained in hot spring water and concentration difference of $CO_2$ in the incubator (Hot Spa B).

The results for Figures 17 and 18 are shown in Table 3. This table shows the rate of increase in $CO_2$ concentration against free carbon dioxide contained in hot spring water installed in the incubator. It showed the average value and standard deviation value for each amount of hot spring water (i.e., 500 mL, 1000 mL, and 2000 mL) and the whole. These values were obtained for the concentration difference in $CO_2$ in the incubator divided by the concentration difference in free carbon dioxide contained in the hot spring water. In other words, it shows the amount of the $CO_2$ concentration increment in the incubator for

1 ppm of free carbon dioxide contained in hot spring water. The difference concentration of $CO_2$ in the incubator was indicated as air (dif), and the difference concentration of free carbon dioxide contained in the hot spring water was indicated as free (dif). This table also shows the results of converting 500 mL to 1000 mL and converting 2000 mL to 1000 mL. The conversion to 1000 mL was calculated by doubling the value for 500 mL and halving the value for 2000 mL. For the air (dif)/free (dif) of Hot Spa A, the range indicated showed a large difference, ranging from 4.7 to 16.2. The 1000 mL equivalent resulted in a smaller range of 7.5 to 9.3. For the air (dif)/free (dif) of Hot Spa B, the range was 5.7 to 11.3, with a larger range of 4.6 to 11.4 in terms of 1000 mL. One possible reason why Hot Spa B did not show the same results as Hot Spa A could be the different velocities at which the hot spring water is discharged to the surface. Although the flow velocity was not measured, the flow velocity was faster for Hot Spa B at the time of sampling. The high velocity of the flow may have stirred up the hot spring water and increased the amount of $CO_2$ emitted into the atmosphere before the experiment was conducted. Since the degree of this change depends on the amount of water, the extent of Hot Spa B was considered to have increased. This air (dif)/free (dif) indicated how much $CO_2$ will be produced by 1 ppm free carbon dioxide. For Hot Spa A, there is a possibility that there was some relationship between the amount of water and the concentration converted to 1000 mL, since the range of concentration to be converted became smaller when converted to 1000 mL.

**Table 3.** Average and standard deviation for concentration difference of $CO_2$ in the incubator (air (dif)) per concentration difference of free carbon dioxide contained in hot spring water (free (dif)) (Hot Spa A and Hot Spa B).

| Amount of Water | Average Standard Deviation | Hot Spa A | | Hot Spa B | |
|---|---|---|---|---|---|
| | | Air (dif)/Free (dif) | Air (dif)/Free (dif) 1000 mL Conversion | Air (dif)/Free (dif) | Air (dif)/Free (dif) 1000 mL Conversion |
| 500 mL (8, 6) * | average | 4.7 | 9.3 | 5.7 | 11.4 |
| | standard deviation | 1.3 | 2.7 | 2.1 | 4.2 |
| 1000 mL (8, 8) * | average | 7.5 | 7.5 | 4.6 | 4.6 |
| | standard deviation | 2.5 | 2.5 | 2.3 | 2.3 |
| 2000 mL (6, 3) * | average | 16.2 | 8.1 | 11.3 | 5.6 |
| | standard deviation | 4.4 | 2.2 | 5.1 | 2.5 |
| whole (22, 17) * | average | 8.9 | 8.3 | 6.2 | 7.2 |
| | standard deviation | 5.5 | 2.5 | 3.6 | 4.4 |

\* Sample number (left: Hot Spa A, right: Hot Spa B). air (dif): The difference concentration of $CO_2$ in the incubator. free (dif): The difference concentration of free carbon dioxide.

From the above, Figures 19 and 20 were created based on the relationship between air (dif) and free (dif) in Table 3. Hot Spa A in Figure 19 showed y = 0.0078X + 0.34 (coefficient of determination; $R^2$ = 0.98). It meant that air (dif)/free (dif) increased in proportion to the amount of hot spring water. Hot Spa B in Figure 20 showed y = 0.0041X + 2.4(coefficient of determination; $R^2$ = 0.55). Although the coefficient of determination was 0.55, the value was low when the amount of hot spring water was 1000 mL. Therefore, it was hard to prove that there was a correlation for Hot Spa B.

The 1000 mL conversion of air (dif)/free (dif) was 8.3 ± 2.5 for the whole value of Hot Spa A. This meant that it was possible to increase $CO_2$ in the incubator by 8.3 ± 2.5 ppm per 1 ppm of free carbon dioxide contained in hot spring water. There was a peculiar result, and it showed a high value of 31.3 in August 2021. The air (dif)/free (dif) was 9.3 ± 5.4 in taking this value into account. This value of 31.3 was excluded as an outlier in this paper because it was above 25.5 (i.e., average + standard deviation × 3). The 1000 mL conversion of air (dif)/free (dif) for Hot Spa B was 7.2 ± 4.4 for the whole value. Hot Spa A and Hot Spa B can increase in $CO_2$ concentration in the incubator by 8.3 ± 2.5 ppm or 7.2 ± 4.4 per 1 ppm of free carbon dioxide contained in hot spring water.

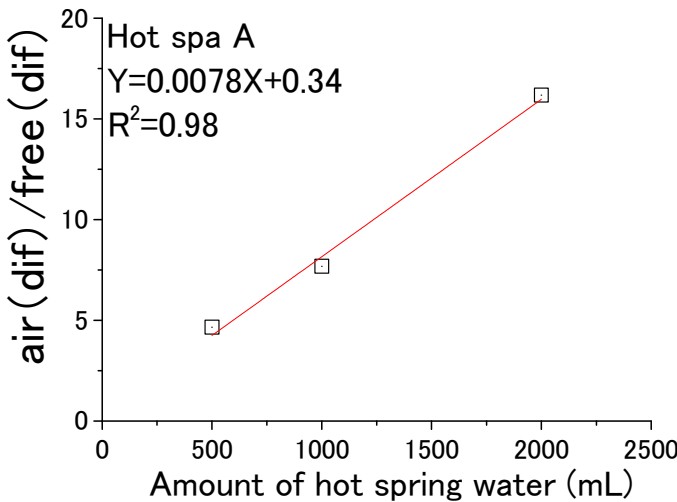

**Figure 19.** Air (dif) per free (dif) for amount of hot spring water (Hot Spa A).

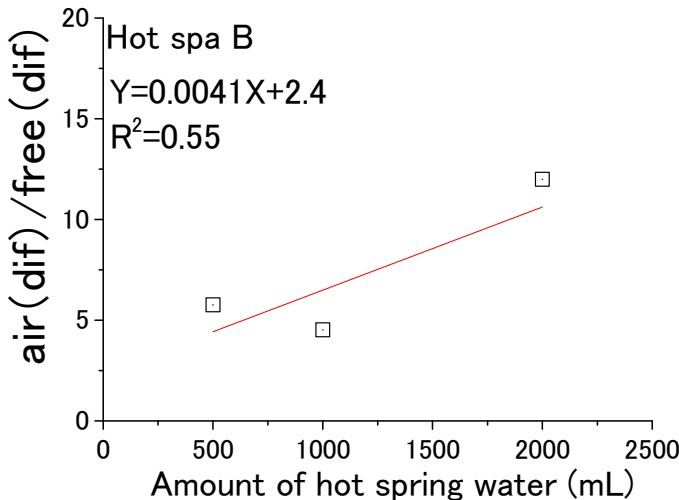

**Figure 20.** Air (dif) per free (dif) for amount of hot spring water (Hot Spa B).

*3.6. Proposals for Utilization the Hot Spring Water as Carbon Dioxide Spring in Facility Horticulture and Plant Factories*

The amount of increase for the $CO_2$ concentration in the air may increase more the larger the contact area between the air and hot spring water is. It showed the same trend with 500 mL and 1000 mL for the relation between the concentration difference of free carbon dioxide contained in hot spring water and $CO_2$ concentration difference in the incubator, as shown Figures 17 and 18. In the case of 2000 mL, it showed a high value compared with 500 mL and 1000 mL when the concentration difference of free carbon dioxide was the same value. The largest contact area with air was the experimental container for the amount of 2000 mL, and 500 mL and 1000 mL were the same area and smaller than the 2000 mL container. It was possible that more $CO_2$ in the air could be increased due to the larger the contact area. It is better to increase the contact area between the hot spring water and the air for increasing the concentration of $CO_2$ in the greenhouse.

The volume inside the incubator was $2.17 \times 10^7$ mm$^3$. The increasing concentration of $CO_2$ in the air was $8.3 \pm 2.5$ (Hot Spa A) and $7.2 \pm 4.4$ (Hot Spa B) for per 1 ppm of free carbon dioxide in 1000 mL of hot spring water. It was possible to obtain the necessary $CO_2$ concentration by changing the amount of hot spring water introduced. The concentration in the greenhouse could be further increased when all free carbon dioxide becomes the $CO_2$ in the incubator. For example, if there was 100 ppm of the free carbon dioxide contained

in the hot spring water, then the $CO_2$ concentration in the greenhouse was increased 580–1080 ppm (calculation formula: $(8.3 - 2.5) \times 100$ ppm to $(8.5 + 2.5) \times 100$ ppm). It was possible to increase the concentration from 290 to 540 ppm after circulating the air when the volume of a target facility or plant factory was twice as large as the incubator. If this amount of the $CO_2$ increment was not enough, it could be applied by increasing the amount of hot spring water in the facility because increasing the amount of hot spring water increased the emission amount into the air for 1 ppm of free carbon dioxide, as is shown in the results in Figures 19 and 20. In the case of Hot Spa A, the increase to 1 ppm free carbon dioxide in 2000 mL of water was $16.2 \pm 4.4$. It meant it could be increased from 1180 to 2060 ppm for 100 ppm. In this case of twice the volume of the incubator, it was possible that the $CO_2$ concentration could be increased by 590–1030 ppm. A volume 1000 times larger would increase the $CO_2$ concentration by 1.180 to 2.060 ppm; however, it was a very small amount. Therefore, it will use these findings of this study. For example, if a facility wants to increase the $CO_2$ concentration by 500 ppm, multiply the rate of increase in the concentration of free carbon dioxide by the rate of increase in the amount of hot spring water, which should be 242.7 to 423.7 times. As an example calculation, with three times the concentration of free carbon dioxide (300 ppm), by increasing the amount of hot spring water by a factor of 80.9 to 141 (162 to 282 L), the target concentration of 500 ppm could be increased.

From the above, the emission of $CO_2$ from carbonated hot spring water can be controlled by the amount of hot spring water. Hot spring water containing free carbon dioxide can be used after taking the volume of greenhouse horticulture and the required $CO_2$ concentration of plants into consideration. Furthermore, it is possible to use the application in cold/warm areas and high/low humidity areas. When used in such a facility, $CO_2$ concentration will be able to be further increased by release from the soil, although it may be affected by the partial pressure of $CO_2$ in a facility. If these facilities can be set up to constantly circulate fresh spa water, there is no particular need for the commonly used ventilation to supply $CO_2$. Furthermore, the characteristics of hot spring water can be used to increase humidity and temperature, making it fully applicable even in closed environments where there is no circulation with the outside air with a ventilation fan. As an example of its use, it would be very useful when growing plants that are concerned about introducing pathogens from outside air. This can be an alternative to the case of using $CO_2$ cylinder gas at the time of seeding because of the effects of pathogens. Based on the above, it can be expected that the hot spring water targeted in this study will grow more plants in a sustainable manner. Furthermore, carbonated hot springs exist throughout the world and may be similar to the hot spring water in this study. There are high expectations for using those hot spring waters to promote plant growth.

It is recommended to install water channels to increase the contact area between greenhouse and hot spring water, in addition to promoting stirring in the facility where hot spring water is introduced.

## 4. Conclusions

This study examined whether it was possible to increase the $CO_2$ concentration in the greenhouse by the emission of free carbon dioxide from volcanic hot spring water by using the incubation method. It was to clarify the effect for increasing of $CO_2$ concentration by the implementation time in the demonstration experiment, temperature and humidity in an incubator, and hot spring water volume for the emission amount of free carbon dioxide. Based on the above results, it was to examine the possibility of applying free carbon dioxide contained in hot spring water to a greenhouse horticulture and a plant factory. Field experiments used the incubator with internal dimensions of 305 mm $\times$ 285 mm $\times$ 250 mm (i.e., volume of $2.17 \times 10^7$ mm$^3$). Experiments were conducted using the incubator method to check only the influence from hot spring water and to remove the influence of $CO_2$ from the outside air and soil. It was possible to increase the concentration of $CO_2$ in the greenhouse using free carbon dioxide contained in hot spring water. It was found that the

implementation time and temperature and humidity in the incubator did not affect the increasing of $CO_2$ concentration. The amount of hot spring water was an impact factor for the increase in $CO_2$ concentration. The results for target hot spring waters showed it was possible to increase $CO_2$ in the incubator by $8.3 \pm 2.5$ ppm or $7.2 \pm 4.4$ per 1 ppm of free carbon dioxide contained in hot spring water. These rising concentrations were a proportional relationship with the amount of hot spring water. In other words, increasing the amount of water also means increasing the rate of increase in $CO_2$ concentration. By increasing the area of contact between the hot spring water and the atmosphere, the amount of $CO_2$ emitted from the hot spring water increased. Although it was not possible to show a relationship by contact area, the results showed that the area of 2000 mL emitted more $CO_2$ than the area of 500 mL or 1000 mL.

From the above, the free carbon dioxide contained in volcanic hot spring water could be put to practical use in greenhouse horticulture and a plant factory. As a method for its practical use, the hot spring water would be replaced every few minutes and water channel with a large area for contact between air and hot spring water would be created.

**Funding:** This research was funded by Takahashi Industrial and Economic Research Foundation in Japan.

**Data Availability Statement:** All of the research data published in this report is owned by Yokota Lab in National Institute of Technology, Wakayama College.

**Conflicts of Interest:** The author declares no conflict of interest.

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
