# Peer review of "Utilization of Hot-Spring-Water-Bound CO2 for Horticulture Plants Using Incubation Method"

_sustainability, doi:10.3390/su151612504_

Round 1

Reviewer 1 Report (Previous Reviewer 1)

Authors studied the utilization CO2 from hot spring water for growing horticulture plants from carbonated spring at Nagayu Spa in Japan. It is very good idea for the researchers to utilize CO2. Results are presented well. Few recommendations I have suggested before acceptance.

1. Title do not present actual work that you studied. Change the tile to “Utilization of hot springs water bound CO2 for horticulture plants using (?) Method”. In parenthesis use appropriate name of CO2 extraction method from hot springs.

2. Atmosphere is a wide term use appropriate term for surroundings where plants are growing. It may be greenhouse etc.

3. Give appropriate name to the method that you use whether it was ventilation or incubation method. It will more useful to boost your research in wide readership.

4. Rewrite the whole abstract by adding important findings.

5. In section 2.2.3. remove “how to” just use measurement of pH,…. etc.

6. In results discussion, discuss each figure separately with respective fig number. Especially from fig 5 to 8. You mixed-up the discussions.

7. Need author’s special attention on grammatical mistakes in language.

Need author’s special attention on grammatical mistakes in language.

Author Response

1-1. Title do not present actual work that you studied. Change the tile to “Utilization of hot springs water bound CO2 for horticulture plants using (?) Method”. In parenthesis use appropriate name of CO2 extraction method from hot springs.

Answer)

We have changed the title as you indicated. We have adopted the incubation method for that title.

After the change)

Utilization of hot springs water bound CO2 for horticulture plants using incubation method

1-2. Atmosphere is a wide term use appropriate term for surroundings where plants are growing. It may be greenhouse etc.

Answer)

The atmosphere inside the incubator was primarily used as the greenhouse, while the general atmosphere was used as the atmosphere. Therefore, greenhouse was used in the sentence indicating the purpose and abstract.

1-3. Give appropriate name to the method that you use whether it was ventilation or incubation method. It will more useful to boost your research in wide readership.

Answer)

I used the incubation method as you suggested.

1-4. Rewrite the whole abstract by adding important findings.

Answer)

As you indicated, I have added new findings to the abstract. The added findings and revisions were as follows.

I added that the hot spring water was influenced by volcanoes, changed atmosphere to greenhouse, that hot spring water was available if volume was taken into account, and that the culture method was used.

1-5. In section 2.2.3. remove “how to” just use measurement of pH,…. etc.

Answer)

We have corrected it as you indicated.

Before the change)

How to measure pH, EC, ORP, water temperature and dissolve components

After the change)

Measurements of pH, EC, ORP, water temperature and dissolve components

1-6. In results discussion, discuss each figure separately with respective fig number. Especially from fig 5 to 8. You mixed-up the discussions.

Answer)

Each figure was discussed separately in its own right.

1-7. Need author’s special attention on grammatical mistakes in language.

Answer)

I have rechecked the grammar as you suggested.

Reviewer 2 Report (Previous Reviewer 3)

It has been improved and can be accepted.

Author Response

It has been improved and can be accepted.

Comment)

Thank you very much for your confirmation.

Reviewer 3 Report (Previous Reviewer 2)

The authors failed to revise according to reviewer's suggestion, why do you submit it again before all are done? Please check line by line for reviewer's comments in this journal, as otherwise, reject again.

Same as last version, the research method is too easy.

Japanese remains in the Table.

Poor English, what does it mean by free carbon dioxide?

What are free and not free carbon dioxide?

State academic, practical and policy contributions of this paper.

State orginality of this paper.

The research method is too easy.

The author said "The further method of supplying CO2, Plants are grown in horticultural agriculture by reusing CO2 emitted when Boilers burn" They should also add that Plant biodiversity increases soil organic carbon storage, according to the paper  Achieving carbon neutrality in China: Spectral clustering analysis of plant diversity

Within the paper, please state originality of this paper.

Introduce all sections of this paper in introduction and research question in this section.

Why "Hot spring water transported by pipes from the source was collected directly into plastic sampling containers. A plastic graduated cylinder was used when sampling the specified amount of water" in 2.2.1?

Missing citation for 2.2.2 and many others.

Line 174, what is DKK-TOA Corp., Tokyo, Japan, model number: HM-30?

And what is DIONEX, America, Commonwealth of Massachusetts, model number: ICS- 1000?

2.2.4 missing citation

shorten 3.2 in title.

What is JIS K 0102:2013, Japan standard?

Where is hot spring water in this paper? Why that was chosen? Can the results be generalised?

Polish english

Author Response

3-1 Same as last version, the research method is too easy.

Answer)

As stated in the objectives, we were not sure if carbonic acid could really be emitted from the spa water.

Therefore, we used this method, although it is a simple one, to conduct the study.

In addition, the experiment was conducted without fixing various conditions in order to approximate the emission under natural conditions. Based on the above, a simple experiment was conducted.

3-2 Japanese remains in the Table.

Answer)

As you indicated, we have changed Japanese to English.

3-3 Poor English, what does it mean by free carbon dioxide?

Answer)

Free carbon dioxide means H2CO3 (CO2+H2O). Therefore I have added H2CO3 at the first word of free carbon dioxide.

3-4 What are free and not free carbon dioxide?

Answer)

Free carbon dioxide means H2CO3 (CO2+H2O). Therefore I have added H2CO3 at the first word of free carbon dioxide.

3-5 State academic, practical and policy contributions of this paper.

Answer)

They were listed in L128-131 and L132-134 of the Introduction. They are as follows.

The academic and practical of this paper were on the free carbon dioxide contained in volcanic hot spring water and examine whether it is possible to increase the concentration of CO2 in the atmosphere.

The policy contributions were to examine the possibility of applying free carbon dioxide in hot spring water to greenhouse horticulture and plant factories.

3-6 State orginality of this paper.

Answer)

The originality is the use of hot spring water, which contains a high level of carbonation but does not produce bubbles like carbonated water.

This was described in L126-128.

3-7 The research method is too easy.

Answer)

This would be the same as the answer to 3-1, we were not sure if carbonic acid could really be emitted from the spa water. Therefore, we used this method, although it is a simple one, to conduct the study.

In addition, the experiment was conducted without fixing various conditions in order to approximate the emission under natural conditions. Based on the above, a simple experiment was conducted.

3-8 The author said "The further method of supplying CO2, Plants are grown in horticultural agriculture by reusing CO2 emitted when Boilers burn" They should also add that Plant biodiversity increases soil organic carbon storage, according to the paper  Achieving carbon neutrality in China: Spectral clustering analysis of plant diversity

Answer)

I have added the following information regarding organic carbonic acid storage as you indicated.

Applicable Sentences)L53-56 in introduction

organic carbonate soil is the largest terrestrial carbon reservoir for the supply of CO2. According to the review, worldwide, soils release about 10 times more greenhouse gases compared to fossil fuel combustion [38]. The use of these gas is suitable from a carbon-neutral perspective.

3-9 Within the paper, please state originality of this paper.

Answer)

The originality is the use of hot spring water, which contains a high level of carbonation but does not produce bubbles like carbonated water.

This was described in L126-128.

3-10 Introduce all sections of this paper in introduction and research question in this section.

Answer)

In the Introduction, we have presented the general content of our discussion in this paper. Here is the text

Applicable Sentences)L132-138 in introduction

This study focused on the free carbon dioxide contained in volcanic hot spring water and examine whether it is possible to increase the concentration of CO2 in the atmosphere. It was examined to clarify the free carbon dioxide emitting performance by the implementation time in demonstration experiment, temperature and humidity in the incubator, and hot spring water volume. The purpose of this study is to examine the possibility of applying free carbon dioxide in hot spring water to greenhouse horticulture and plant factories.

3-11 Why "Hot spring water transported by pipes from the source was collected directly into plastic sampling containers. A plastic graduated cylinder was used when sampling the specified amount of water" in 2.2.1?

Answer)

The reason was difficult to collect water directly from the source. The contents of this report have been added.

Applicable Sentences)L160-162 in 2.2.1

The reason for using transported hot spring water was that it was difficult to extract water directly from the source.

3-12 Missing citation for 2.2.2 and many others.

Answer)

I have added the citation as you indicated it was not available.

Applicable Sentences)L165-166 in 2.2.2

Free carbon dioxide was determined by titration with sodium hydroxide in accordance with Japan Industrial standard (JIS K 0102:2013, Japan Industrial standard) [67]

3-13 Line 174, what is DKK-TOA Corp., Tokyo, Japan, model number: HM-30?

And what is DIONEX, America, Commonwealth of Massachusetts, model number: ICS- 1000?

Answer)

We have included this information because, as is the default in Susy(MDPI), the equipment is supposed to include the name of the company, the location of the company's headquarters, and the model number.

3-14 2.2.4 missing citation

Answer)

The description is based on the equipment manual, so we have added that information. However No citation is provided as we are not discussing the issue using citations.

Applicable Sentences)L203-204 in 2.2.2

According to the manual, the performance of the equipment is as follows.

3-15 shorten 3.2 in title.

Answer)

Before change)

“3.2. Relation between free carbon dioxide contain in hot spring water and carbon dioxide in the incubator for concentration change on experiment”

After change)

“3.2. Relation between free carbon dioxide and carbon dioxide for the concentration change”

3-16 What is JIS K 0102:2013, Japan standard?

Answer)

JIS K 0102 means Japan Industrial standard. A word was omitted and has been corrected.

Applicable Sentences)L166 in 2.2.2

(JIS K 0102:2013, Japan Industrial standard) [67]

3-17 Where is hot spring water in this paper? Why that was chosen? Can the results be generalised?

Answer)

Hot Spa A and Hot Spa B spa waters were used in this study. The words connecting them were not found in the text, but only in the explanatory part of the figure, so we have added them.

The reason for selecting these waters is that they contain high concentrations of carbonic acid and minerals. This sentence has already been mentioned. Also, as mentioned in the introduction, there are carbonated springs in many countries, and we think that the results of this study can be used when using these springs. We have added this point as well.

Applicable Sentences)L141-143 in 2.2.2

The demonstration experiment was conducted using hot spa A and hot spa B of Nagayu Spa (Japanase name is Nagayu Onsen). Their hot spring waters located in Taketa City, Oita Prefecture in JAPAN as shown in Figure.1.

Applicable Sentences)L148-151 in 2.2.2

Such hot springs exist throughout the world as indicated in the“1.intoroduction”. Therefore, the results of this study could be applied to those hot spring waters.

Round 2

Reviewer 3 Report (Previous Reviewer 2)

Introduction

·       The introduction is quite long at over 2 pages. I would suggest condensing some of the background information to focus more on the motivation and objectives of this specific study. The general information about CO2 and hot springs could be shortened. Research questions about the paper should be added, introduce the later few sections in the end of introduction.

·       Some additional citations may be needed in the introduction to better tie this work to previous literature. For example, when discussing the benefits of higher CO2 concentrations for plant growth, citing some prior studies would strengthen this claim.

·       The originality and significance of this particular study using the incubation method could be highlighted more clearly early in the introduction.

Methods

·       The figures font size should be standardized per journal requirements.

·       More details are needed on why the specific hot spring waters were chosen and how they were collected. This information is currently lacking.

·       The methods section could benefit from more citations to standard procedures, equipment, etc. For example, citing methodology standards for the free CO2 measurements. Citing some previous research also use regression per paper “Achieving Carbon Neutrality – The Role of Heterogeneous Environmental Regulations on Urban Green Innovation”

·       Parts of the methods need clarification. For example, the purpose and setup of the incubator experiments could be explained more clearly.

Results/Discussion

·       In the results, consider adding more interpretation and discussion of the data trends and mechanisms. The results are currently presented without much analysis.

·       The figures could be improved by ensuring axes are properly labeled, units are indicated, and captions are detailed enough for the main takeaways.

·       Some of the conclusions in the discussion go beyond what the data fully supports. The conclusions need to be tightened to align with the evidence presented.

Overall

·      The paper could benefit from careful editing for grammar, wording, and clarity throughout. There are some awkward phrases. The paper should revise according to previous reviews' comments on top of this one before resubmitting again. Many of the previous comments were ignored and in fact that was the reason of reject in last submission.

·       Some more discussions on recent CO2 research should be added, say, As of 22 April 2022, 83 countries worldwide have set their goals to reduce global carbon emissions by 74% etc perA Study on Public Perceptions of Carbon Neutrality in China: has the Idea of ESG Been Encompassed?”

·       The implications and applications of this work for horticulture could be expanded on more in the conclusions.

·       The referencing style needs to be made consistent throughout the paper.

·       The title and abstract are appropriate but could highlight the main findings on the incubation method's effects more clearly.

Polish English

Author Response

Comments and Suggestions for Authors

Introduction

[Indication 1-1]

  • The introduction is quite long at over 2 pages. I would suggest condensing some of the background information to focus more on the motivation and objectives of this specific study. The general information about CO2 and hot springs could be shortened. Research questions about the paper should be added, introduce the later few sections in the end of introduction.

Answer)

I have removed a large portion of the introduction. I added another section you mentioned, but I kept it to no more than two pages.

change point)

These are the part of the introduction that states “Delate”.

[Indication 1-2]

  • Some additional citations may be needed in the introduction to better tie this work to previous literature. For example, when discussing the benefits of higher CO2 concentrations for plant growth, citing some prior studies would strengthen this claim.

Answer)

I cited prior studies.

change point)L34-37

In several previous studies, plant growth was enhanced by increasing the concentration of CO2. Also, elevated atmospheric CO2 will increase nitrogen uptake, and continued nutrient supply will sustain long-term growth [16-22].

[Indication 1-3]

  • The originality and significance of this particular study using the incubation method could be highlighted more clearly early in the introduction.

Answer)

The originality and importance of this particular study using the incubation method was emphasized more clearly earlier in the introduction.

change point)L69-73

Thus, plant growth has been studied using carbonated springs. In these studies, a quantitative study for the emitting of free carbon dioxide and CO2 in carbonated spring water is needed. It is especially important to validate the experimental method without the influences of CO2 supply from boilers, soils and CO2 in the atmosphere.

Methods

[Indication 1-4]

  • The figures font size should be standardized per journal requirements.

Answer)

The size of each figure was changed to standardize the font size of the figures and tables were also changed to a form that allows direct input.

[Indication 1-5]

  • More details are needed on why the specific hot spring waters were chosen and how they were collected. This information is currently lacking.

Answer)

It was not listed as you indicated, so I have added it.

change point)L131-135

These hot spring waters were selected from among the many sources within Nagayu Hot Springs because preliminary survey conducted by our laboratory showed high CO2 concentrations within Nagayu Hot Springs and they could be freely collected for a survey. The preliminary survey was conducted to investigate the concentration of bicarbonate ions (HCO3-).

[Indication 1-6]

  • The methods section could benefit from more citations to standard procedures, equipment, etc. For example, citing methodology standards for the free CO2 measurements. Citing some previous research also use regression per paper “Achieving Carbon Neutrality – The Role of Heterogeneous Environmental Regulations on Urban Green Innovation”

Answer)

Explanation of standard methods of measuring free carbon dioxide and citation of previous studies using regression were provided.

change point)L158-162, L243-245

There are several methods for measuring free carbonic acid, and this method is one of them and is the most commonly used method for on-site measurement. Other methods include analysis using a TOC meter performed in a laboratory. Free carbonic acid has the characteristic of being released over time. Therefore, it is difficult to measure the original concentration with this method.

Although it is necessary to examine each factor comprehensively using regression analysis for what needs to be resolved as in the references, this study first examined each of the targeted factors one by one to check their effects [49].

[Indication 1-7]

  • Parts of the methods need clarification. For example, the purpose and setup of the incubator experiments could be explained more clearly.

Answer)

The purpose and setup of the experiment in the incubator are clearly described.

change point)L195-199, L678-680

The reason for using an incubator is to conduct the experiment under conditions that minimize the influence of CO2 from the outside air. Plastic greenhouses and other facilities would not be able to completely block the entry of outside air and could also be affected by soil and other factors. An incubator is also equipped with a fan, which can be used to generate air circulation.

Experiments were conducted using the incubator method to remove the influence of CO2 from the outside air and soil, and to check only the influence from hot spring water.

Results/Discussion

[Indication 1-8]

  • In the results, consider adding more interpretation and discussion of the data trends and mechanisms. The results are currently presented without much analysis.

Answer)

Interpretations and discussions of trends and mechanisms were added to all data.

change point)L298-301, L333-339, L348-352, L356-358, L403-406, L507-515, L538-552

ORP showed a negative value, indicating that the subject hot spring water is in a reduced state. Since much of a subsurface is a reducing environment with poor oxygen, it is possible that it has existed underground for an unknown period of time and has been affected in some way underground.

When the amount of hot spring water was 2 or 3 liters, free carbon dioxide remained in the hot spring water even when the experiment was terminated. When the amount of hot spring water was 1 L, it showed a phenol check and the concentration was very low. In addition for 1L, the amount of CO2 concentration in the incubator increased by a small amount. This means that CO2 release will continue until the amount of free carbon dioxide is very low.

This factor can be assumed to be due to the air in the incubator being circulated and stirred by the fan, eliminating the shading of CO2 concentration. As mentioned in the introduction, it can be inferred from previous studies that without the use of fans to circulate air in a greenhouse, there will be shading in the concentration of CO2.

This was thought to be due to the fact that when the amount of hot spring water was low, the amount of free carbon dioxide contained was also low.

The reason for the greater emit at the beginning of the experiment was that the partial pressure of the hot spring water was higher, making it easier for CO2 to be emitted. It could be assumed that as time passes, the partial pressure difference with the air in the incubator became smaller and less likely to be emitted.

Based on the above, the coefficient of determination results showed a relationship for the same amount of water. For the different amount of waters, the slope was almost the same for Hot Spa A, except for the significantly outlier values. However, the intercepts yielded different results. Therefore, it was a possible that the amount of water has some effect on the intercepts. Therefore, it was necessary to consider what trends would be observed if converted to the same amount of water. In addition to the amount of hot spring water, the concentration of free carbon dioxide may have an effect on the results, so it was necessary to consider the trend per 1 ppm of free carbon dioxide.

For the air(dif)/ free(dif) of Hot Spa A, The range indicated showed a large difference, ranging from 4.7 to 16.2. The 1000 mL equivalent resulted in a smaller range of 7.5 to 9.3. For the air(dif)/ free(dif) of Hot Spa B, the range was 5.7 to 11.3, with a larger range of 4.6 to 11.4 in terms of 1000 mL. One possible reason why Hot Spa B did not show such results as Hot Spa A was that the velocity at which these hot spring waters were discharged to the ground were different. Although the flow velocity was not measured, the flow velocity was faster for Hot spa B at the time of sampling. The high velocity of the flow may have stirred up the hot spring water and increased the amount of CO2 emitted into the atmosphere before the experiment was conducted. Since the degree of this change depends on the amount of water, the extent of Hot spa B was considered to have increased. This air(dif)/free(dif) indicated how much CO2 will be produced by 1 ppm free carbon dioxide. For Hot spa A, there is a possibility that there was some relationship between the amount of water and the concentration converted to 1000mL, since the range of concentration to be converted became smaller when converted to 1000 mL.

[Indication 1-9]

  • The figures could be improved by ensuring axes are properly labeled, units are indicated, and captions are detailed enough for the main takeaways.

Answer)

Axis reorganization and detailed caption descriptions have been added.

change point)

Figure.2, Table.1, Table.2, fugure.5 to 8

[Indication 1-10]

  • Some of the conclusions in the discussion go beyond what the data fully supports. The conclusions need to be tightened to align with the evidence presented.

Answer)

The statement "stirring is good" was removed because it is beyond the range supported by the data. Also, an explanation was added to support the conclusion regarding contact area.

change point)L688-692

The amount of CO2 emitted tended to increase as the contact area between hot spring water and the atmosphere was increased. Although it was not possible to show a relationship by contact area, the results showed that the area of 2000 mL emitted more CO2 than the area of 500 mL or 1000 mL.

Overall

[Indication 1-11]

  • The paper could benefit from careful editing for grammar, wording, and clarity throughout. There are some awkward phrases. The paper should revise according to previous reviews' comments on top of this one before resubmitting again. Many of the previous comments were ignored and in fact that was the reason of reject in last submission.

Answer)

I am very sorry. I could not understand the meaning of your comment.

I have added the detailed explanation of originality and the significance of this study.

[Indication 1-12]

  • Some more discussions on recent CO2 research should be added, say, As of 22 April 2022, 83 countries worldwide have set their goals to reduce global carbon emissions by 74% etc per “A Study on Public Perceptions of Carbon Neutrality in China: has the Idea of ESG Been Encompassed?”

Answer)

I added recent trends in CO2 research.

change point)L52-57

According to a report by the IPCC, 48% reduction in global carbon dioxide emissions in 2030, 65% reduction in 2035, and 80% by 2040 are needed to prevent a "1.5 degree Celsius temperature increase from pre-industrial times [36]. Other CO2 supply methods, such as biogas, require more time to release CO2. The best method is one that can supply CO2 as quickly as a ventilation fan and does not require even many resources.

[Indication 1-13]

  • The implications and applications of this work for horticulture could be expanded on more in the conclusions.

Answer)

The implications and applications of this work to horticulture is detailed in the conclusion.

change point)L639-647, L652-665

A volume 1000 times larger would increase CO2 concentration by 1.180 to 2.060 ppm, however it was a very small amount. Therefore, it will use these findings of this study. For example, if you want to increase the CO2 concentration by 500 ppm, multiply the rate of increase in the concentration of free carbon dioxide by the rate of increase in the amount of hot spring water, which should be 242.7 to 423.7 times. As an example of calculation, by three times the concentration of free carbon dioxide (300 ppm) and increasing the amount of hot spring water by a factor of 80.9 to 141 (162 to 282 L), the target concentration of 500 ppm could be increased.

When used in such a facility, CO2 concentration will be able to be further increased by release from the soil, although it may be affected by the partial pressure of CO2 in the facility. If facilities can be set up to constantly circulate fresh spa water, there is no particular need for the commonly used ventilation to supply CO2. Furthermore, the characteristics of hot spring water can be used to increase humidity and temperature, making it fully applicable even in closed environments where there is no circulation with the outside air by a ventilation fan. As an example of its use, it would be very useful when growing plants that are concerned about introducing pathogens from the outside air. This can be an alternative to the case of using CO2 cylinder gas at the time of seedling because of the effects of pathogens. Based on the above, it can be expected that the hot spring water targeted in this study will grow more plants in a sustainable manner. Furthermore, carbonated hot springs exist throughout the world and may be similar to the hot spring water in this study. There are high expectations for using those hot spring waters to promote plant growth.

[Indication 1-14]

  • The referencing style needs to be made consistent throughout the paper.

Answer)

The method of describing references has been standardized based on the format.

[Indication 1-15]

  • The title and abstract are appropriate but could highlight the main findings on the incubation method's effects more clearly.

Answer)

The same as 1-7, the effect of the incubation method is described.

change point)L195-199

The reason for using an incubator is to conduct the experiment under conditions that minimize the influence of CO2 from the outside air. Plastic greenhouses and other facilities would not be able to completely block the entry of outside air and could also be affected by soil and other factors. An incubator is also equipped with a fan, which can be used to generate air circulation.

Round 3

Reviewer 3 Report (Previous Reviewer 2)

Review: A lot of the previous comments were ignored. The authors must revise according to the reviewers comments.

  1. Originality & Relationship to Literature: The paper presents a unique approach to utilizing free carbon dioxide (H2CO3) from hot springs water for horticulture. The concept is novel and contributes to the field of sustainable horticulture. However, the literature review could be more comprehensive. More recent studies on the use of volcanic CO2 for similar purposes are missing (e.g. [37, 38]). This could help to better situate the study in the context of existing research.

  2. Some related authoritive journal article should be cited How does fiscal decentralization affect CO2 emissions? The roles of institutions and human capital

    Energy Economics 94, 105060

  3. Results & Conclusions: The results are not clearly presented and analyzed due to the truncation of the paper. Consequently, it's also not possible to comment on the conclusions drawn.

  4. Implications for research, practice, and/or society: The paper has not clearly identified the implications of the research for practice or society. Drawing out these implications would enhance the paper's value.

  5. Quality of Communication: The paper is generally clear and concise, but there are several grammatical errors that need to be addressed. For example, in the phrase "Plants are grown in horticultural agriculture by reusing CO2 emitted when Boilers burned," the word "Boilers" should not be capitalized and "burned" should be "are burned."

  6. Miscellaneous: The introduction seems to be within an acceptable length. However, there is no clear research question stated in the introduction, which makes it challenging to understand the paper's purpose. Moreover, the citation format does not seem to adhere to any recognized standard, and this makes it difficult to track the sources used.

Overall, it needs significant improvements in terms of literature review, clarity in research questions, grammatical accuracy, and citation format. Also, it's recommended to include more sections of the paper for a comprehensive review.

Polish English.

Author Response

Response to previous review.

Indication 1-3

  • The originality and significance of this particular study using the incubation method could be highlighted more clearly early in the introduction.

Indication 1-7

  • Parts of the methods need clarification. For example, the purpose and setup of the incubator experiments could be explained more clearly.

3-7 The research method is too easy.

Answer

I have reexamined the text and indicated the importance of the matter reviewer pointed out in the previous section. I briefly presented it in the introduction and detailed the experimental method in 2.2.4.

Added text and locationL75-79、L211-234

In these studies, a quantitative study for the emitting of free carbon dioxide and CO2 in carbonated spring water is needed. It is especially important to validate the experimental method without the influences of CO2 supply from boilers, soils and CO2 from the atmosphere.

The general method of experiments using an incubation involves constant temperature through the operation of heaters and air circulation through the operation of fans. In this study, in order to confirm the amount of free carbon dioxide emitted from hot spring waters under natural conditions, the experiment was conducted with only fans running and no heaters [45]. The months of the experiment were in August, September, October, December, 2021 and March, 2022.

The experiment used the incubator as shown in figure.2. According to the manual, the performances of the equipment were as follows. The shape and size of the incubator were Square prism with effective internal dimensions of 305 x 285 x 250 mm. The volume was 2.17 x 107 mm3. The small size reduces the difference of CO2 concentration in the incubator and is a good way to check the effects of the emitting from hot spring water. One of the problems in managing CO2 in horticultural facilities is that the large volume of the facility and inadequate circulation within the facility can cause differences in CO2 concentration within the facility. This causes differences in plant growth. The small size of the system allows for sufficient circulation to accurately measure the amount emitted from hot spring water. Assuming application to actual institutional horticulture, the size of this incubator is a very small volume. Since the concentration of CO2 is expressed in per volume (ppm), its application to horticultural facilities can be determined by calculation if the difference in volume of the facilities is quantitatively clear. The size of the incubator used in the experiment has no significant effect on its application to horticultural facilities.

1.Originality & Relationship to Literature:

The paper presents a unique approach to utilizing free carbon dioxide (H2CO3) from hot springs water for horticulture. The concept is novel and contributes to the field of sustainable horticulture. However, the literature review could be more comprehensive. More recent studies on the use of volcanic CO2 for similar purposes are missing (e.g. [37, 38]). This could help to better situate the study in the context of existing research.

4.Implications for research, practice, and/or society: The paper has not clearly identified the implications of the research for practice or society. Drawing out these implications would enhance the paper's value.

Answer

I have added a few sentences in the Introduction about contributing to the field of sustainable horticulture, making the literature review comprehensive, and Implications for research, practice, and/or society.

In addition, I could not find any recent studies on the use of volcanic CO2 except for the references listed here. The position of this study is explained in terms of the food situation. The additional text is the same as above.

Added text and locationL25-37、L116-124

The global population is growing, and according to the United Nations Population Fund (UNFPA), the world population exceeded 8 billion in November 2022 [1]. Its population is expected to continue to grow and is projected to exceed 9 billion by 2050 [2]. Food supply and demand is expected to increase accordingly. Therefore, in order to secure food, it is necessary to cultivate many plants and further accelerate their growth. However, re-sources for this purpose are skyrocketing worldwide [3]. Therefore, the prices of resource energy and mineral resources used for growing plants are also increasing, making it difficult to secure food. Furthermore, in the context of global warming, emissions of carbon dioxide and other artificial greenhouse gases need to be curbed [4]. This consideration makes it impossible to avoid the economic impact. As a countermeasure, fiscal decentralization has been shown to have the potential to reduce CO2 emissions [5]. From the perspective of the SDGs, plants need to be grown using methods that address these various issues in a sustainable manner [6].

By solving these problems, it can find how CO2 in hot spring water can be used to promote plant growth. This will not require energy resources and will promote plant growth, thus contributing to solving food shortages in the future. This CO2 will be released into the atmosphere if not utilized. If it is utilized, there is a possibility that the CO2 concentration in the atmosphere can be reduced. Furthermore, hot spring water is a resource that can be used sustainably as long as it is not used in the wrong quantities, allowing for sustainable use and sustainable plant growth. In addition, the use of a carbonated hot spring water is expected to have a positive impact on the economy as it reduces the use of energy resources and removes artificially created CO2.

2.Some related authoritive journal article should be cited How does fiscal decentralization affect CO2 emissions? The roles of institutions and human capital Energy Economics 94, 105060

Answer

I have included this information in the introduction that I have added to this issue.

Added text and location))L34-36

This consideration makes it impossible to avoid the economic impact. As a countermeasure, fiscal decentral-ization has been shown to have the potential to reduce CO2 emissions [5].

3.Results & Conclusions: The results are not clearly presented and analyzed due to the truncation of the paper. Consequently, it's also not possible to comment on the conclusions drawn.

Answer

Some of the figures were not fully explained. In particular, there were no caption explanations for Figures 13 and 14, so I have added them.

Added text and locationL329-334、L348-352、L510-522、L578-588

The horizontal axis shows the time of the experiment and the vertical axis shows the CO2 change in the incubator as measured by the CO2 data logger. The experiment in Figure 5 was conducted for the longest time, more than 4 hours, while the rest of the experiments in Figures 6 to 8 were completed within 2 hours. For Figures 6 to 8, the experiment was terminated because the concentration of free carbon dioxide in the hot spring water had decreased.

To measure free carbon dioxide contain the hot spring water, the door was opened to take the container with hot spring water from the incubator. For the analysis of free carbonic acid, the amount of hot spring water necessary for the measurement was taken from the container. The container was then returned to the incubator and the door was completely closed.

As for the air temperature and humidity, they are those in the incubator. Temperature and humidity are increased by placing the hot spring water in the incubator. The experiment is conducted with the fan running and the door closed. About the caption, “Start of experiment” indicates temperature or humidity in the incubator before the experiment as just before placing hot spring water and “End of experiment” indicates temperature or humidity in the incubator after the experiment. Thus, there are two results for one change in CO2 concentration per minute. For example, Figure 13 shows two results in the area of approximately 1100 ppm CO2 concentration, one for the temperature at the start of experiment and the other for the temperature after the end of experiment. After the experiment, the results are concentrated on the right side of the figure due to higher temperatures and humidity. Although it would be sufficient to show only the results after the experiment, the results before the experiment are also shown.

Therefore, the difference in concentration of free carbon dioxide in the hot spring water is the concentration of free carbon dioxide measured at the beginning of the experiment mi-nus the concentration of free carbon dioxide measured contain to take the hot spring water out from the incubator after the emitting experiment in the incubator was completed (e.g. if the concentration at experiment starts is 500 mg/L and ends is 100 mg/L, the value subtracted is 400 mg/l.). The difference in CO2 concentration in the incubator is also calculated by subtracting the value measured by the CO2 data logger at the beginning of the experiment from the value when the spa water is taken after the emitting experiment in the incubator is completed (For example, if 400 ppm was measured at the start of the experiment and 1000 ppm at the end, the value subtracted would be 600 ppm.).

5.Quality of Communication: The paper is generally clear and concise, but there are several grammatical errors that need to be addressed. For example, in the phrase "Plants are grown in horticultural agriculture by reusing CO2 emitted when Boilers burned," the word "Boilers" should not be capitalized and "burned" should be "are burned."

Answer

In addition to the areas you pointed out, we reconfirmed the entire project.

6.Miscellaneous: The introduction seems to be within an acceptable length. However, there is no clear research question stated in the introduction, which makes it challenging to understand the paper's purpose. Moreover, the citation format does not seem to adhere to any recognized standard, and this makes it difficult to track the sources used.

Answer

The significance of this study has been added to the introduction, which is the same as the first and fourth sections.

I have also created a bibliography based on the template below. In addition, I have added DOIs so that you can track them.

[Template of sustainability]

1.Author 1, A.B.; Author 2, C.D. Title of the article. Abbreviated Journal Name Year, Volume, page range.

2.Author 1, A.; Author 2, B. Title of the chapter. In Book Title, 2nd ed.; Editor 1, A., Editor 2, B., Eds.; Publisher: Publisher Location, Country, 2007; Volume 3, pp. 154–196.

3.Author 1, A.; Author 2, B. Book Title, 3rd ed.; Publisher: Publisher Location, Country, 2008; pp. 154–196.

4.Author 1, A.B.; Author 2, C. Title of Unpublished Work. Abbreviated Journal Name year, phrase indicating stage of publication (submitted; accepted; in press).

5.Author 1, A.B. (University, City, State, Country); Author 2, C. (Institute, City, State, Country). Personal communication, 2012.

6.Author 1, A.B.; Author 2, C.D.; Author 3, E.F. Title of Presentation. In Proceedings of the Name of the Conference, Location of Conference, Country, Date of Conference (Day Month Year).

7.Author 1, A.B. Title of Thesis. Level of Thesis, Degree-Granting University, Location of University, Date of Completion.

8.Title of Site. Available online: URL (accessed on Day Month Year).

Overall, it needs significant improvements in terms of literature review, clarity in research questions, grammatical accuracy, and citation format. Also, it's recommended to include more sections of the paper for a comprehensive review.

Answer

As noted above, I have made significant revisions to the literature review, clarity of research questions, and citation format. Grammar has also been checked again.

This manuscript is a resubmission of an earlier submission. The following is a list of the peer review reports and author responses from that submission.

Round 1

Reviewer 1 Report

Authors studied the increase in atmospheric carbon dioxide by using free carbon dioxide contained in the carbonated spring at Nagayu Spa in JAPAN. This study contained many non-scientific aspects. Introduction is very short, irrelevant and do not support the topic. Lack of supportive literature is the major drawback of the study. Standard methods and experimental procedures are not supported by literature. Figures are nicely presented but discussions of the results still needs more description and literature support. Current topic will not attract wide readership.

Abstract need to add important results. CO2 is formula not an abrivation no need to write “Carbon dioxide is abbreviated as CO2 below” (Line 28).

English language required editing.

Reviewer 2 Report

The abstract has not yet incorporate the research gaps that it tries to fill, originality, academic, practical and policy contributions.

In abstract, exact dimensions like 2.17 × 107 mm3 are of no value to reader, but it will be best to state like whether the volume used is of particular size (larger than or smaller than ordinary ones) and its implications to the research results.

The introduction part has to include originality, research question and research gaps.

Lines 52-58, sections 2.2-2.3 missing citation.

State the source of the map, it may needs to seek for copyright.

Section 3, extend the 3 sentences before 3.1.

How did the authors obtain figures 3 and 4?

Describe before Tables 1 and 2.

Setting of the experiment like r. The amount of hot spring water was 1000 171 mL, 2000 mL and 3000 mL by using 1000 mL vessel (i.e. 159 x 124 x 80 mm) and 2000 mL 172 vessel (i.e. 276 x 152 x 93 mm; it can be insert over 3000mL). The time intervals for meas- 173 uring the free carbon acid were indeterminately. The operation way of the fan was dif- 174 ferent depends on the experiment as shown in figure. Phenol check in Figure.8 means 175 that the concentration of free carbon dioxide was not 0mg/L, but it was very small con- 176 centration has to be explained and provide some explaination based on cited journals.

Figure 6 and 11 need to remove the Japanese (?) word.

Figures need to be redrawn.

Why simple one variable regression like Figure 13 is good enough? Most are affected by multiple factors.

Even multiple regression may not be good enough in many cases.

The author should expand to close to 30 references.

Polish english

Reviewer 3 Report

I found it very interesting and would like to suggest the following improvement to be done by the authors.

1. Throughout the manuscript either carbon dioxide or CO2 should be used.

2. The last paragraph of the introduction, "This study focused on the free carbon dioxide contained in hot spring water and examine whether it is possible to increase the concentration of CO2 in the atmosphere."

authors should discuss this point.......why and its advantages......??

3 Some of the figures have not been discussed properly in the text. Please discuss properly.

4. Heading 3 and 4 should be merged together to form 'Result and Discussion'

5. Grammatical mistakes need to be checked and improved throughout the manuscript. 

simple grammatical mistakes need to be checked/improved.